# Spatiotemporal dynamics of signal dependent exocytosis and parasitophorous vacuolar membrane rupture during *Plasmodium falciparum* egress

**Watcharatip Dedkhad**[1,2], **Tia Tran**[1,2¤a], **Manuel A. Fierro**[1,2¤b], **Carrie Brooks**[1], **Vasant Muralidharan**[1,2]*

1 Center for Tropical and Emerging Global Diseases, Athens, Georgia, United States of America,
2 Department of Cellular Biology, University of Georgia, Athens, Georgia, United States of America

¤a Current address: Saint Louis University School of Medicine, Saint Louis, Missouri, United States of America
¤b Current address: Department of Genetics and Biochemistry, Eukaryotic Pathogen Innovation Center, Clemson University, Clemson, South Carolina, United States of America
* vasant@uga.edu

## Abstract

Malaria, caused by intracellular *Plasmodium falciparum* parasites, remains a major global health concern. These parasites reside and replicate within a vacuole in host red blood cells. Egress of daughter parasites out of the vacuolar and host membranes is tightly regulated via a complex mechanism. Prior studies have suggested that a cyclic-GMP driven calcium signaling pathway leads to the signal-dependent exocytosis of egress-specific vesicles that discharge several proteases into the parasitophorous vacuole. However, signal-dependent exocytosis during egress has not yet been observed in live parasitized RBCs. We targeted the exocytosis reporter, superecliptic pHlourin or SEP, to these egress-specific vesicles and utilized live imaging to observe exocytosis. The spatiotemporal relationship between exocytosis and the breakdown of the parasitophorous vacuolar membrane (PVM) as well as parasite egress was also determined using a fluorescent reporter fused to EXP2. Our data showed that exocytosis is triggered as early as 3 hours prior to merozoite egress. These data suggest that the PVM rupture occurs at a single site and rapidly expands from that initial site of rupture, releasing the merozoites into the RBC. This is followed by RBC membrane rupture and egress of merozoites. Using conditional mutants of *Plasmodium* endoplasmic reticulum calcium-binding protein (PfERC), we demonstrate that loss of PfERC inhibits signal-dependent exocytosis of egress-specific vesicles. Together, these data demonstrate that signal-dependent exocytosis of egress-specific vesicles starts well before merozoites are formed via cytokinesis, PVM ruptures at a single site, and that PfERC is required for exocytosis of egress-specific vesicles.

which permits unrestricted use, distribution, and reproduction in any medium, provided the original author and source are credited.

**Data availability statement:** All relevant data are within the manuscript and its Supporting Information files.

**Funding:** This work was supported by the National Institutes of Health (R56AI173133 and R01AI179950 to VM; T32AI060546 to MAF). The funders had no role in study design, data collection and analysis, decision to publish, or preparation of the manuscript.

**Competing interests:** The authors have declared that no competing interests exist.

## Author summary

Malaria is a deadly disease caused by intracellular Plasmodium falciparum parasites that grow and divide within a vacuole inside human red blood cells (RBCs). At the end of its lifecycle within RBCs, the parasite makes several daughter cells that need to egress from the infected RBCs continue their lifecycle. How and when the parasite decides to exit the host RBC is currently unclear. Here, we used live video microscopy coupled with fluorescent reporters to directly observe several key processes during parasite egress. Our data suggest that the parasite prepares for egress several hours before the actual event by secreting vesicles containing proteins that facilitate egress. We also observed that the membrane that surrounds the daughter parasites also breaks in only one spot, though what determines that spot and how that membrane ruptures remain to be determined. We also found that a calcium binding protein functions in the secretion of the egress specific vesicles. Our study suggests that there may be two separate triggers for parasite egress that may or may not be linked and provides new insights into the mechanisms driving egress that may be targeted for antimalarial drug development.

## Introduction

Malaria is a life-threatening disease caused by parasites in the genus *Plasmodium,* and infection by *P. falciparum* causes the majority of the estimated mortality of nearly 600,000 in 2024 [1]. Clinical malaria manifestations of malaria result from the proliferation of *Plasmodium* parasites in human red blood cells (RBCs). *P. falciparum* parasites replicate via schizogony to produce up to 32 mature daughter parasites roughly every 48 hours. Parasite growth and replication occurs within a single membrane vacuole known as the parasitophorous vacuole (PV) within the host RBC. The PV provides a niche essential for parasite asexual development and replication by schizogony. At the end of the schizogony, the mature daughter parasites must rupture the PV membrane (PVM) and the host RBC membrane to exit or egress from the current host and invade a new RBC to continue their asexual expansion.

Egress of mature daughter parasites or merozoites is a tightly regulated complex multistep process, triggered by poorly understood signaling pathway(s) [2–11]. Egress is triggered by an unknown signal that leads to the production of cyclic GMP (cGMP), which activates Protein Kinase G (PKG) that leads to the release of intracellular calcium via unknown mechanisms [11,12]. The rise in intracellular calcium then triggers the exocytosis of egress-specific vesicles known as exonemes as well as invasion-specific vesicles known as micronemes [11,12]. Exonemes contain two key proteases, subtilisin 1 (SUB1) and plasmepsin X (PMX) that function in egress of merozoites [10,13–15]. The release of SUB1 and PMX into the PV starts a proteolytic cascade that eventually leads to breakdown of the PVM and RBC membrane [10,13–16]. However, the spatiotemporal regulation of exocytosis, PVM breakdown, and merozoite egress remains to be fully understood.

Most prior studies utilized immunofluorescence assays using fixed parasites to observe exocytosis, a dynamic process. These studies have suggested that exoneme exocytosis is upstream of PVM rupture and occurs during the final minutes before egress [8,17,18]. Our current model for *P. falciparum* egress suggests this signaling pathway occurs in the final 10 minutes of the intraerythrocytic asexual lifecycle to release merozoites from membranes [19]. This model, based on inhibitor-based studies, suggest that PKG signaling causes exocytosis of exonemes triggering a protease cascade in the PV that leads to rounding of the PVM followed by rupture of the PVM that irreversibly leads to merozoite egress [8,11,16,19]. However, since exoneme exocytosis has never been observed in live parasites, the spatiotemporal regulation of this process as well as its relationship to PVM rupture remains unknown. Furthermore, all previous studies observe microneme exocytosis, as a proxy for exoneme exocytosis, in fixed parasites [12]. However, micronemes do not function in egress and contain proteins required for merozoite invasion. This raises several questions such as: 1) When does exoneme exocytosis begin prior to merozoite egress? 2) What is the spatiotemporal relationship between exocytosis and PVM rupture? 3) What molecular mechanisms are required for signal-dependent exocytosis? To answer these questions, we wanted to directly observe exoneme exocytosis and PVM rupture in live parasites to refine our model of *Plasmodium* egress and test the role of specific proteins in these processes.

In this study, we first adapted a reporter system to observe exoneme exocytosis in live parasites during egress. The changes in PVM morphology during merozoite egress was observed using a fluorescent marker targeted to the PVM [19]. We generated parasite lines that expressed the exocytosis reporter in exonemes and a fluorescent marker on the PVM. Using these dual-labeled parasites, we observed exoneme exocytosis and PVM rupture simultaneously in *P. falciparum* infected RBCs undergoing egress. This allowed us to establish the spatiotemporal relationship between exoneme exocytosis, PVM rupture, and merozoite egress. Finally, to determine the role of PfERC in exoneme exocytosis, we generated PfERC conditional mutants expressing the exoneme-targeted exocytosis reporter. Our results show that exoneme exocytosis occurs several hours prior to egress and that PfERC is required for the signal-dependent exocytosis of exonemes.

## Results

### Targeting an exocytosis reporter to exonemes

To observe exoneme exocytosis in live parasites, we utilized a pH-sensitive variant of green fluorescent protein (GFP) known as superecliptic pHluorin or SEP [20,21] that is widely used to monitor the exocytosis and pH of subcellular compartments in several organisms including *P. falciparum* [22,23]. SEP fluorescence is quenched in acidic environments (pH < 6) and is fluorescent in neutral pH environments [20,21,24,25] (Fig 1A). Secretory vesicles are acidified as they are generated and trafficked from the Golgi. Thus, we hypothesized that SEP will not be fluorescent within acidic secretory vesicles, like exonemes, but will fluoresce when exonemes fuse to the plasma membrane and SEP is released into the neutral pH environment of the PV during exocytosis (Fig 1A).

Our current models for *P. falciparum* egress suggest that exoneme exocytosis occurs in the final minutes of egress [12,19]. To determine the relationship between exoneme exocytosis and PVM rupture, we wanted to simultaneously monitor both these processes in live parasites. Therefore, we used CRISPR/Cas9 gene editing to generate a double-labeled parasite mutant, where we tagged exported protein 2 (EXP2), a protein located in the PVM with mRuby3 (PfEXP2^mRuby3) [26] and the exoneme protein, PMX [10,13–15], with SEP (PfEXP2^mRuby3/PMX^SEP; Fig 1B and 1C). These data show that we successfully tagged both endogenous EXP2 and PMX gene loci with mRuby3 and SEP, respectively (Fig 1C and 1D). Since PMX is essential for the completion of the intraerythrocytic lifecycle, these data also suggest that fusion of SEP to PMX did not inhibit its biological function (Figs 1C, 1D and S1). The expression of EXP2 protein fused with mRuby3 and PMX fused with SEP was detected by western blot and showed the expected sizes, including the correct maturation species for PMX (Fig 1D). Live imaging of late schizonts showed PMX^sep fluorescence as it was likely secreted into the PV, delineated by the EXP2^mRbuy3 labeled PVM (Fig 1E), indicating the tagging did not change the subcellular localization of either protein. Together, these data showed that tagging the *pmx* locus with SEP did not affect its maturation or function.

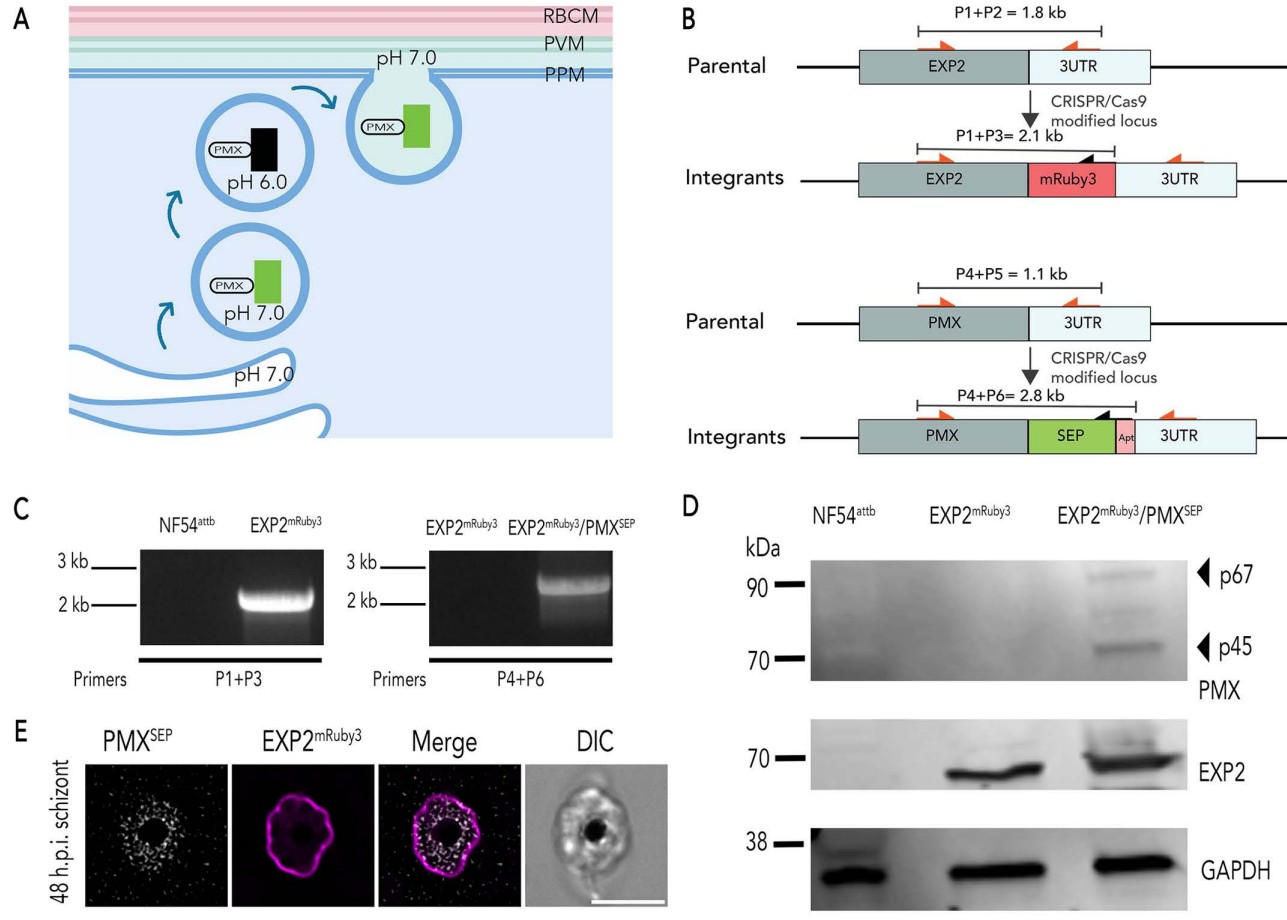

**Fig 1. Dual-fluorescent NF54ᵃᵗᵗᴮ parasites to monitor exoneme exocytosis and PVM rupture.** (A) The schematic shows Plasmepsin X (PMX) tagged with super-ecliptic pHluorin or SEP. SEP is non-fluorescent at pH 6.0 or less, but it is fluorescent at pH 7. (B) Schematics of the targeting (pMGPMX-SEP; Top) and (pbPM2-EXP2-mRuby3; bottom) plasmids, the expected recombination product, and representation of the primer pairs used to analyze an integration. (C) PCR diagnosis of mRuby3 fusion to the PfEXP2 locus in *P. falciparum* NF54ᵃᵗᵗᴮ (left) and PMXˢᴱᴾ in the PfEXP2ᵐᴿᵘᵇʸ³(right) parasites (primers sequences shown in S1 Table). (D) Western blot of parasite lysates isolated from *P. falciparum* NF54ᵃᵗᵗᴮ, PfEXP2ᵐᴿᵘᵇʸ³, and PfEXP2ᵐ⁻ᴿᵘᵇʸ³/PMXˢᴱᴾ parasites probed with anti-EXP2 and anti-GFP antibodies. Results from one representative experiment of three are shown. (E) A representative image of live PfEXP2ᵐᴿᵘᵇʸ³/PMXˢᴱᴾ schizonts (48 hpi). Images from left to right are PMXˢᴱᴾ (gray), EXP2ᵐᴿᵘᵇʸ³ (magenta), fluorescence merge, and differential interference contrast or phase-contrast (DIC) Scale bar = 5 μm.

## Blocking Protein Kinase G signaling inhibits exoneme exocytosis

Using the PfEXP2ᵐᴿᵘᵇʸ³/PMXˢᴱᴾ parasites, we wanted to observe the spatio-temporal dynamics of exocytosis and PVM rupture. Studies using fixed parasites in immunofluorescence assays have suggested that exocytosis of egress-specific vesicles requires PKG signaling [3,5,9,10]. Therefore, we hypothesized that exoneme exocytosis will be blocked by inhibiting PKG using specific inhibitors like compound 1 (C1) [8]. We incubated synchronized PfEXP2ᵐᴿᵘᵇʸ³/PMXˢᴱᴾ schizonts at 44 hours post-invasion (h.p.i) with 1.5 μM C1. After 4 hours of incubation, the schizonts were washed and either incubated again with or without C1 and observed via live fluorescence microscopy for 30 minutes (Fig 2).

We observed that in the presence of C1, PfEXP2ᵐᴿᵘᵇʸ³/PMXˢᴱᴾ schizonts did not egress during the 30-minute window of observation (Fig 2A and S1 Movie). The PVM was intact in these schizonts during this time period (Fig 2B). These data show that blocking exocytosis via PKG inhibition also inhibits PMXˢᴱᴾ fluorescence (Fig 2B), indicating that the

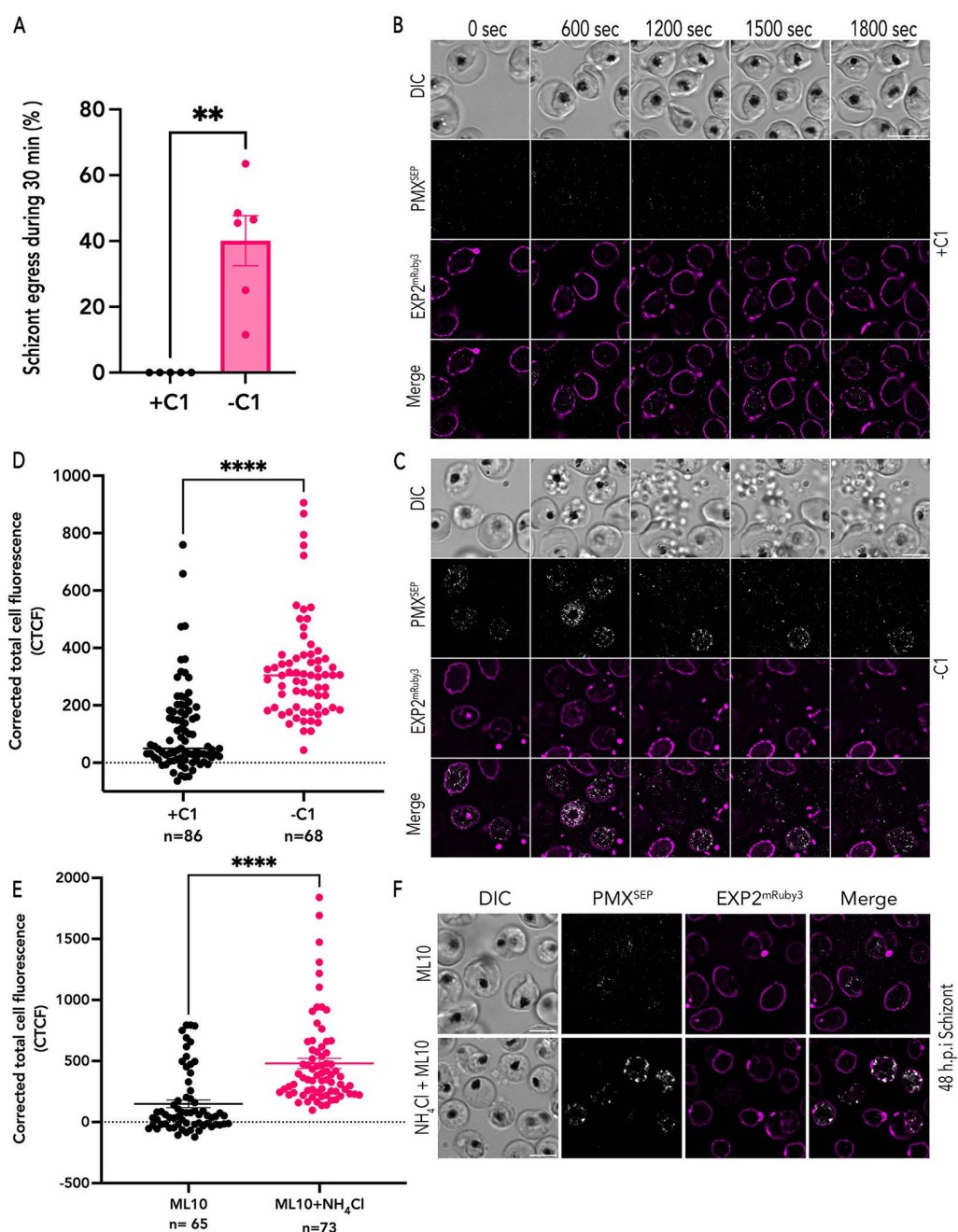

**Fig 2. PKG inhibition blocks exoneme exocytosis.** (A) Quantification of PfEXP2$^{mRuby3}$/PMX$^{SEP}$ parasite egress in the presence of C1 (+C1; black) or absence of C1 (-C1; magenta) during the 1800-second recording period. Each dot represents data from one time-lapse imaging experiment (n = 5 recordings from 3 biological replicates for PfEXP2$^{mRuby3}$/PMX$^{SEP}$ parasites with C1, n = 6 recordings from 3 biological replicates for PfEXP2$^{mRuby3}$/PMX$^{SEP}$ parasites without C1, error bars = SEM; **p < 0.001 by unpaired two-tailed t test) (B) A representative image from live imaging of synchronized PfEXP2$^{mRuby3}$/PMX$^{SEP}$ schizonts in the presence of C1. (C) A representative image from live imaging of synchronized PfEXP2$^{mRuby3}$/PMX$^{SEP}$ schizonts after C1 removal. Parasite egress occurs, and free-merozoites are scattered in the extracellular space (DIC). (D) Corrected total cell fluorescence (CTCF) of PMX$^{SEP}$ quantified from time-lapse images of synchronized PfEXP2$^{mRuby3}$/PMX$^{SEP}$ schizonts incubated with (black; n = 86, 3 biological replicates) or without (magenta; n = 68, 4 biological replicates) C1. ****p-value<0.0001, unpaired t-test. error bars = SEM. (E) Corrected total cell fluorescence (CTCF) of PMX$^{SEP}$ quantified from snapshots of live synchronized PfEXP2$^{mRuby3}$/PMX$^{SEP}$ (48 h.p.i) schizonts incubated with ML10 (black; n = 65, 3 biological replicates) or ML10 and NH$_4$Cl (magenta; n = 73, 3 biological replicates) C1. ****p-value <0.0001, unpaired t-test. error bars = SEM. (F) A representative image from live imaging of synchronized PfEXP2$^{mRuby3}$/PMX$^{SEP}$ schizonts in the ML10 only (Top) and ML10 plus NH$_4$CL (bottom). Scale bar = 5 µm.

environment within unsecreted exonemes is acidic resulting in quenching of SEP fluorescence (Fig 2B). In some schizonts incubated with C1, we observed SEP fluorescence, which may be due to exocytosis of exonemes prior to C1 addition (Fig 2). However, none of these schizonts were able to rupture the PVM or egress during the observation period (Fig 2A).

In the absence of C1, we observed punctate PMX$^{SEP}$ fluorescence within the PVM, indicating PMX$^{SEP}$ was exposed to neutral pH and suggesting that exoneme exocytosis had occurred (Fig 2C and S2 Movie). The time-lapse images are snapshots at a single timepoint and as the vesicle pH is neutralized upon fusion to the parasite plasma membrane, resulting in a punctate PMX$^{SEP}$ fluorescent signal observed from the neutralized exoneme (Fig 2C and S2 Movie). We observe multiple vesicles with PMX being secreted from vesicles into the PV (Fig 2C and S2 Movie). The PMX$^{SEP}$ signal rapidly decreases as it dilutes into the much larger PV compartment. These data suggest that exoneme exocytosis may occur all over the schizont and may not be limited to a single exit site or specific cellular location within the merozoite. However, another possibility is that the vesicles are released from specific locations within the merozoite and the Brownian motion of merozoites results in the appearance that exocytosis occurs at several sites on the membrane. In the absence of C1, exoneme exocytosis occurred first, then the PVM ruptured in most PfEXP2$^{mRuby3}$/PMX$^{SEP}$ schizonts followed by the release of free merozoites (Fig 2C). The corrected total cell fluorescence for each schizont observed via live video microscopy was integrated across several time points (Fig 2D). These data show that in the presence of C1, PMX$^{SEP}$ fluorescence signal is substantially reduced suggesting that exocytosis does not occur when PKG is inhibited (Fig 2D).

Incubating PfEXP2$^{mRuby3}$/PMX$^{SEP}$ schizonts with ML10, another PKG inhibitor [27], also resulted in loss of PMX$^{SEP}$ fluorescence (Figs 2E, 2F and S2). One possibility we considered is that the reduction in PMX$^{SEP}$ fluorescence signal is a result of direct interference from the PKG inhibitors (C1 or ML10). Thus, we aimed to neutralize the acidic pH within exonemes, in the presence of PKG inhibitors, which should also lead to an increase in PMX$^{SEP}$ fluorescence without exocytosis. Ammonium chloride is a weak base that neutralizes the pH in acidic intracellular compartments, including in *P. falciparum* infected RBCs [28,29]. PfEXP2$^{mRuby3}$/PMX$^{SEP}$ schizonts were incubated with ammonium chloride and ML10 showed increased PMX$^{SEP}$ fluorescence unlike those incubated with only ML10 (Fig 2E and 2F). Thus, these data show that exonemes are acidic secretory vesicles, and that their exocytosis is dependent upon PKG signaling.

## Spatiotemporal relationship between exoneme exocytosis and PVM rupture

The spatiotemporal relationship between exoneme exocytosis and merozoite egress has been determined based on studies that utilized fixed parasites and live imaging [12,30]; however, live imaging with specific genetic expression of proteins targeting exoneme and PVM is needed. More recently, live imaging approaches have been utilized to investigate the morphology of the PVM during egress [11,19]. These data show that a calcium-dependent signal causes the rupture of the PVM followed by RBC membrane breakdown to release the merozoites [11,19]. However, a central part of the egress model, the signal-dependent exocytosis of exonemes has not yet been observed by live microscopy. Therefore, we wanted to define the relationship between signal-dependent exocytosis of exonemes and PVM rupture.

To determine the relationship between signal-dependent exocytosis of exonemes and PVM rupture, synchronized PfEXP2$^{mRuby3}$/PMX$^{SEP}$ schizonts (44 h.p.i) were prepared as described previously and incubated with C1 for 4 hours. After removing C1, we observed the schizonts via time-lapse imaging for 30 mins with 30 sec intervals (Fig 3 and S3 Movie). As others have reported [11,19], we observed irregularly shaped PVM at the start of our observation (Fig 3A). About 8 minutes prior to egress, the PVM adopted a rounded shape (Fig 3) and this was followed rapidly (~1 minute) by the first observable break in the PVM (Fig 3B). The schizonts egressed about 7 minutes after the PVM rupture (Fig 3C). Our observations agree with prior studies that also utilized reversible PKG inhibitors to establish the timeline of egress starting from PVM rounding [11,19]. Thus, despite differences in parasite lines and preparation across labs, as well as microscopy settings, the observed timeline of the egress cascade is consistent.

Since our images were taken using 30 sec imaging intervals, we wanted to observe the exact temporal relationship between PVM rounding and PVM rupture as well as observe PVM rupture at a higher temporal resolution. Therefore, we

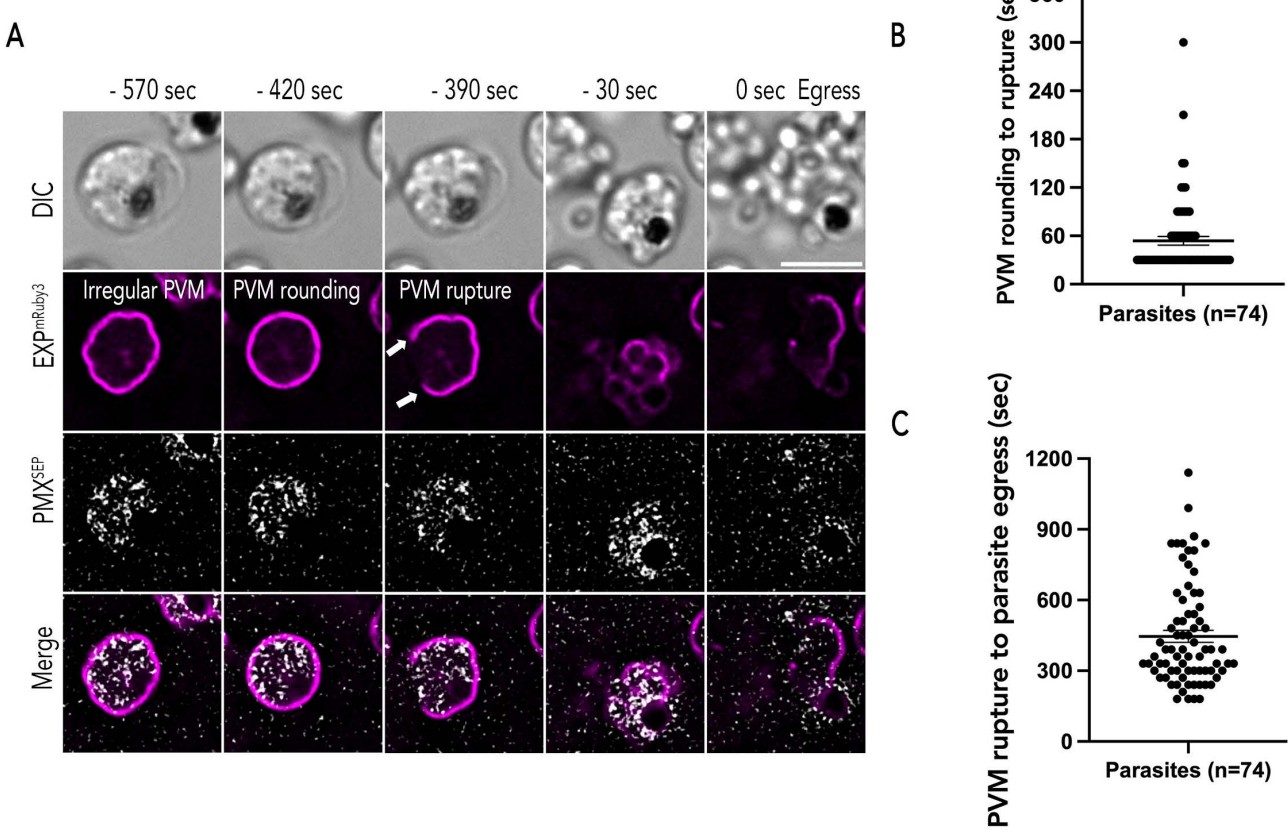

**Fig 3. Exoneme exocytosis prior to egress.** (A) Representative time lapse images (captured at 30 sec intervals) during egress of PfEXP2mRuby3/PMXSEP merozoites. Minus time indicates time points before merozoite egress. PVM (magenta) morphology changes from an irregular shape to a rounded shape. In most schizonts, PVM rupture (white arrows) occurred in the next frame or two (30 to 60 sec) after PVM rounding. Merozoites free from membranes indicate egress. Several exoneme exocytosis (gray) events were observed from the start of the recording until merozoite egress. Scale bar = 5 μm. (B) The estimated time between PVM rounding to rupture (Mean ± SEM = 53.919 ± 5.321 sec) and (C) the time between PVM rupture to merozoite egress (Mean ± SEM = 445.946 ± 25.276 sec). n = 74, 7 biological replicates, double-labeled parasites, error bars = SEM, time-lapse recordings of parasite egress is a 30-sec interval.

utilized 4 sec and 2 sec time-lapse imaging to observe egressing PfEXP2mRuby3/PMXSEP schizonts (Fig 4). Similar to our prior observations (Fig 3), we observed that the PVM morphology changed from an irregular shape to a more rounded shape (Figs 4A, 4B, and S3A-S3C and S4 and S5 Movies). The diameter (~4.3 μm) and circumference (~13.6 μm) of the PVM in all rounded schizonts were similar across biological replicates (Fig 4B and 4C).

Within a few seconds (~25 sec) after the schizont rounds, the PVM ruptures (Fig 4D), though we observed some variation in the time taken for PVM rupture after rounding (Fig 4D). In some schizonts, it took nearly a minute to observe PVM rupture after rounding, while in other schizonts PVM rupture occurred a few seconds (<10 sec) post-rounding (Fig 4D). These data support the hypothesis that PVM rounding is a critical morphological step for PVM rupture to occur (Fig 4D) [11,19]. Regardless of the time interval of imaging, PVM rupture always occurred at a single site after rounding (Fig 4). The size of this initial rupture was consistent across several biological replicates (~2.8 μm; Figs 4E and S3D-S3F). We never observed a second site of PVM rupture during egress and the initial rupture widened in subsequent time-lapse images allowing the merozoites to escape the PVM (Fig 4E). These data suggest that the PVM rupture may be regulated and likely occurs at a specific site on the membrane (Fig 4).

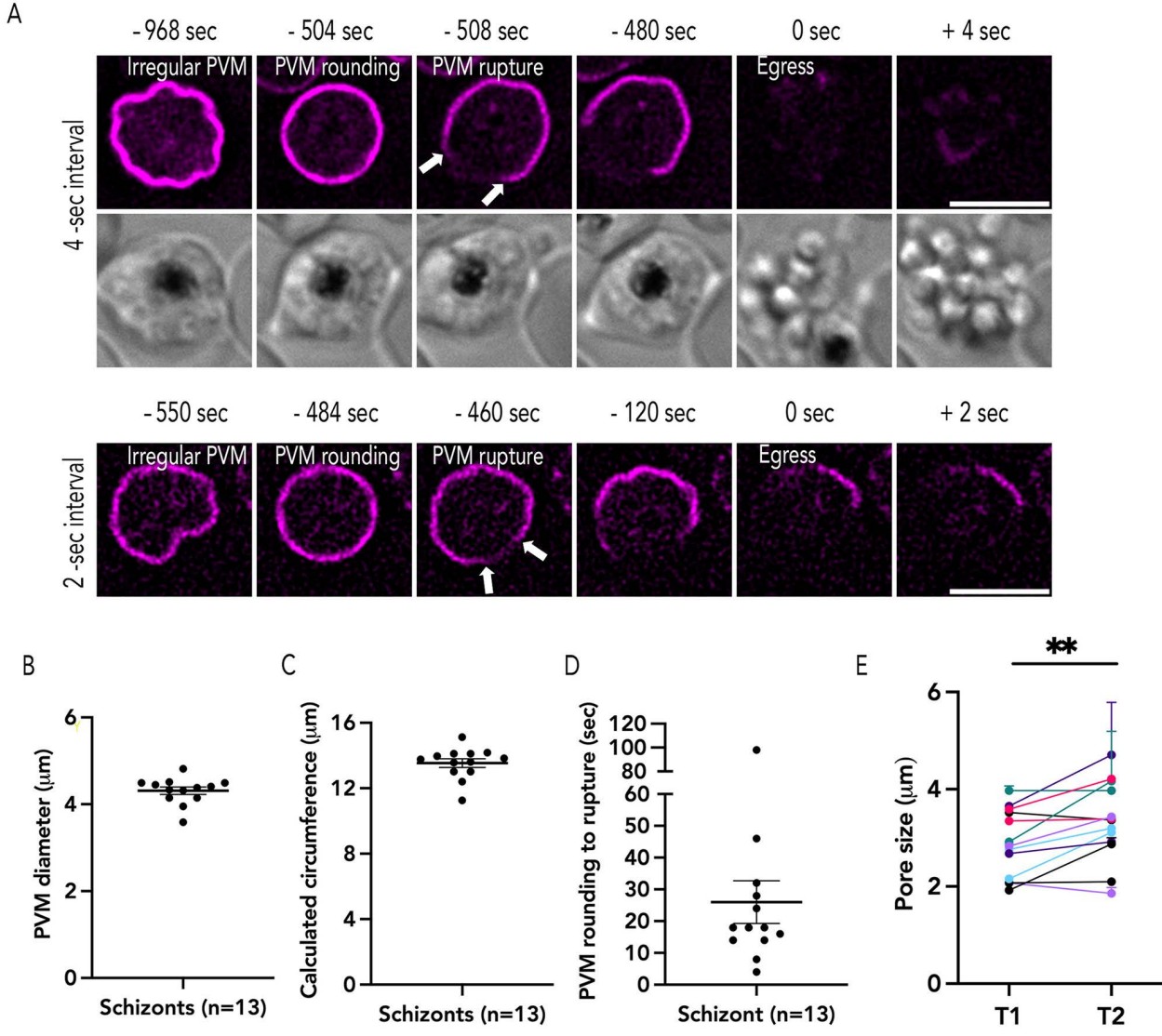

**Fig 4. PVM rupture occurs at a single site.** (A) Representative time lapse images captured from 4-sec (top) and 2-sec intervals (bottom), the white arrow indicates two ruptured ends of the PVM (B) The diameter of rounded PVM from 2-sec interval time lapse images (Mean±SEM=4.31±0.08 μm), (C) circumference (Mean±SEM=13.54±0.264 μm), (D) the duration of PVM rounding (Mean±SEM=26±6.706 sec; n = 13, 4 biological repli-cates). (E) The pore size changes between the first time point PVM breakdown (T1), Mean±SEM=2.88±0.191 μm, and the second time point (T2), Mean±SEM=3.33±0.226 μm (each color represents one schizont imaged every 2 sec, n = 13, 4 biological replicates, error bars = ± SEM, paired t-test with ** p-value ≤ 0.001). The measurement data are shown in S3 Fig. Scale bar = 5 μm.

## Exoneme exocytosis begins several hours prior to merozoite egress

Our studies using reversible PKG inhibitors showed that exoneme exocytosis was already occurring in PfEXP2mRuby3/PMXSEP schizonts at the start of our observation (Figs 2-4). Therefore, we wanted to establish a time-line for signal-dependent exoneme exocytosis and study this process without the use of any inhibitors. Using highly synchronized PfEXP2mRuby3/PMXSEP parasites (44 h.p.i), we monitored exoneme exocytosis until merozoites egressed from these schizonts (Fig 5 and S6 Movie).

We observed several naturally egressing PfEXP2$^{mRuby3}$/PMX$^{SEP}$ schizonts over four biological replicates and at the start of the recording we did not observe any PMX$^{SEP}$ fluorescence in any of these schizonts indicating that exoneme exocytosis had not yet started (Fig 5). Parasite viability was assessed using the movement of schizonts within the infected RBC as well as by hemozoin movement within the food vacuole and only viable schizonts that egressed over this time period were analyzed further (Fig 5). Moreover, the changes in PVM morphology as assessed by PfEXP2$^{mRuby3}$ fluorescence was another measure of parasite viability over the observation time period (Fig 5A and S6 Movie).

In all PfEXP2$^{mRuby3}$/PMX$^{SEP}$ schizonts, we observed punctate PMX$^{SEP}$ fluorescence within the PVM starting several hours (~3 hours) prior to merozoite egress (Fig 5B). Once the punctate PMX$^{SEP}$ fluorescence started, these events were

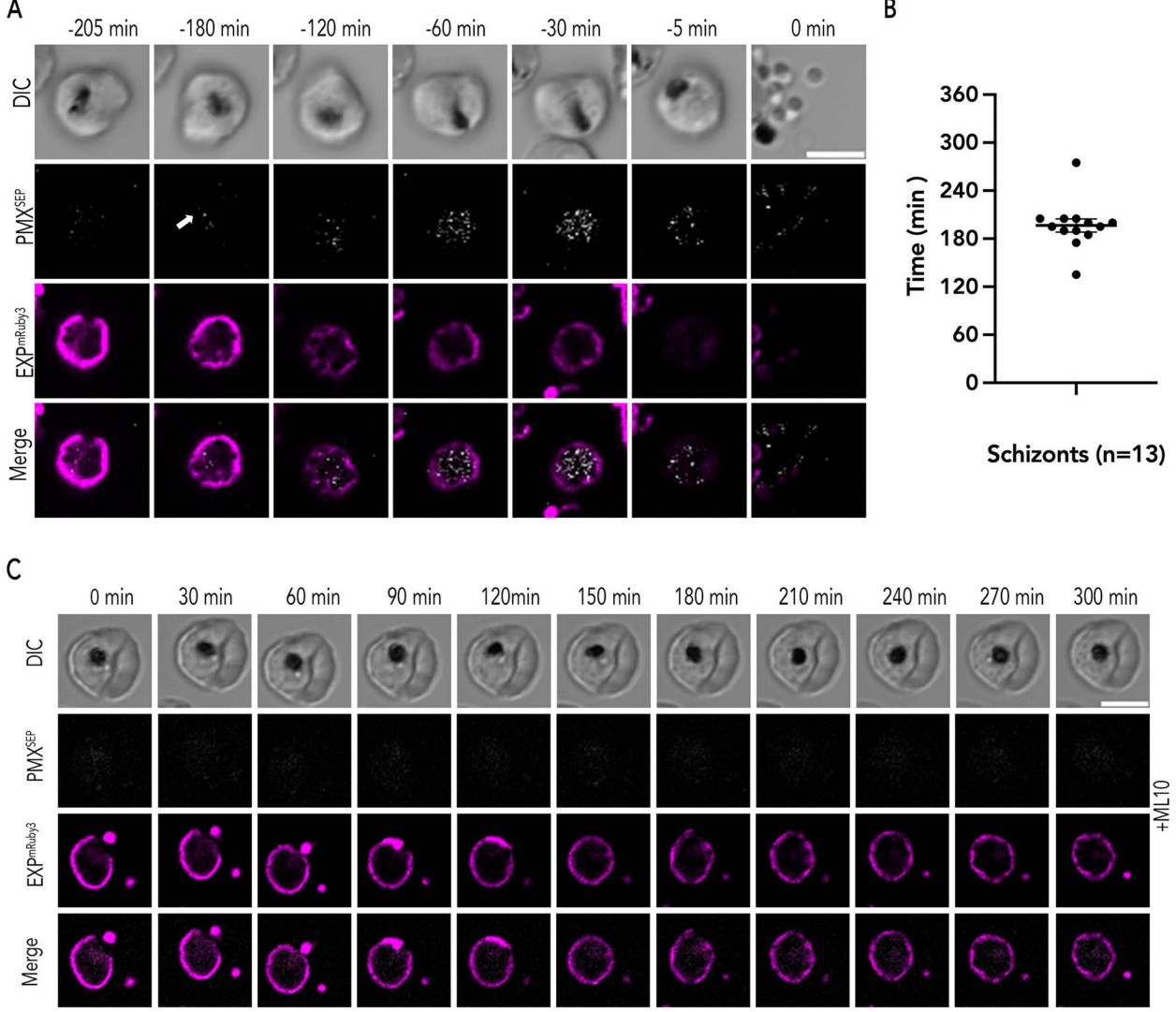

**Fig 5. Exoneme exocytosis begins 3 hours prior to egress.** (A) Representative time lapse images of PfEXP2$^{mRuby3}$/PMX$^{SEP}$ merozoite exocytosis and egress. The arrow indicates the first image where fluorescence signals were observed. Time points prior to merozoite egress (0 min) are indicated. (B) The average time of exoneme exocytosis observed in PfEXP2$^{mRuby3}$/PMX$^{SEP}$ schizonts prior to egress. (n = 13 schizonts from 4 biological replicates, Mean ± SEM = 196.53 ± 8.35 min). (C) Representative time lapse images of PfEXP2$^{mRuby3}$/PMX$^{SEP}$ merozoite in the presence of ML10. Time point shows every 30 min over 5 hours recording. Scale bar = 5 μm.

observed continuously until the merozoites egressed from the infected RBC (Figs 5A and S4A-S4B and S6 Movie). These data suggest that the signals that kickstart exoneme exocytosis occur at least 3 hours prior to egress (Fig 5). To determine if the scale of exocytosis changes during these 3 hours prior to egress or if exocytosis peaks at a specific time before egress, we normalized PMX$^{SEP}$ fluorescence for each schizont (S5 Fig). The normalized PMX$^{SEP}$ fluorescence shows no clear pattern of exocytosis over these 3 hours, with some schizonts showing steady increase in exocytosis until egress (5/13 schizonts), while others show a steady and constant normalized PMX$^{SEP}$ fluorescence until egress (4/13 schizonts) and yet others showing randomly changing PMX$^{SEP}$ fluorescence until egress (4/13 schizonts; S5 Fig). It is possible that gathering more examples may reveal a pattern or that there may not be a clear pattern for exocytosis prior to egress.

To confirm that the PMX$^{SEP}$ fluorescence we observed at 44 h.p.i was due to exoneme exocytosis, we incubated similarly isolated PfEXP2$^{mRuby3}$/PMX$^{SEP}$ schizonts (44 h.p.i) with the PKG inhibitor ML10 over the same time period (Figs 5C and S4C-S4D and S7 Movie). We did not observe any PMX$^{SEP}$ fluorescence suggesting that SEP was already present in the acidic exonemes (Figs 5C and S4C-S4D and S7 Movie). Furthermore, if the PMX is present in acidic vesicles at 44 h.p.i, we reasoned that we should be able neutralize the pH of these vesicles in these schizonts using a weak base like ammonium chloride (Fig 6A-6B). Exocytosis was prevented in these schizonts using the PKG inhibitor, ML10. We observed that PMX$^{SEP}$ fluorescence increased in ammonium chloride-treated parasites compared to untreated schizonts (Fig 6A-6B). Thus, these data demonstrate that PMX$^{SEP}$ is present in mature acidic secretory vesicles at 44 h.p.i, and the observed PMX$^{SEP}$ fluorescence at this time is due to exoneme exocytosis (Fig 5). Together, these data suggest that a critical concentration of egress proteases may be needed in the PV to activate merozoite egress, which may take several hours to achieve (Fig 5). Alternatively, there may be a second signal that activates the egress protease cascade and PVM rounding [11,19].

Since we observed PMX in already acidified vesicles at 44 h.p.i, we wanted to determine when exonemes are acidified during schizogony. The process of vesicle acidification is intrinsic to the secretory pathway and independent of PKG signaling. In this case, PKG inhibitors would not inhibit PMX$^{SEP}$ fluorescence within vesicles that were not yet acidified. Therefore, we assessed PMX$^{SEP}$ fluorescence in 40 h.p.i PfEXP2$^{mRuby3}$/PMX$^{SEP}$ schizonts (Fig 6C and S8 Movie). We found several schizonts that briefly exhibited PMX$^{SEP}$ fluorescence before the signal disappeared (S8 Movie). The loss of PMX$^{SEP}$ fluorescence could either be due to exocytosis of these fluorescent vesicles or because these vesicles were being acidified. As we show (Fig 2) and others have shown [3,5,8,9], vesicle exocytosis depends upon PKG signaling. We reasoned that if the loss of fluorescence was due to vesicle acidification, then inhibiting PKG will have no effect on PMX$^{SEP}$ fluorescence. Therefore, we incubated 40 h.p.i PfEXP2$^{mRuby3}$/PMX$^{SEP}$ schizonts with the PKG inhibitor ML10. As we expected, PMX$^{SEP}$ fluorescence was not inhibited by the PKG inhibitor ML10 (Fig 6D-6E and S9 Movie). Thus, these data show that in 40 h.p.i PfEXP2$^{mRuby3}$/PMX$^{SEP}$ schizonts, vesicles were undergoing acidification. We did not observe any PMX$^{SEP}$ fluorescence in schizonts again until later during schizogony (~44 h.p.i, Fig 5).

A recent study showed that *P. falciparum* V-type ATPases may localize to secretory vesicles in merozoites [31] and therefore, we wanted to determine if these V-type ATPases function to maintain the acidic pH within exonemes. To test this, we incubated PfEXP2$^{mRuby3}$/PMX$^{SEP}$ late schizont with ML10 to prevent exocytosis as well as with the V-type ATPase-specific inhibitor Bafilomycin A1 (BaF). We reasoned that incubation with ML10 will prevent changes in PMX$^{SEP}$ fluorescence due to exocytosis and the addition of BaF will inhibit the V-type ATPase leading to neutralization of the vesicle pH that will result in PMX$^{SEP}$ fluorescence (S6 Fig). No exocytosis or egress was observed in ML10-treated PfEXP2$^{mRuby3}$/PMX$^{SEP}$ parasites (S6A Fig). Surprisingly, we did not see an increasing fluorescence during the 15-min time-lapse videos in ML10-treated PfEXP2$^{mRuby3}$/PMX$^{SEP}$ parasites incubated with BaF (S6B Fig). There was no discernible difference in PMX$^{SEP}$ fluorescence upon incubation with BaF (S6B Fig). These data suggest that the mechanisms maintaining the acidic pH within exonemes are insensitive to BaF (S6 Fig). The data suggest there may be other BaF-insensitive proton pumps or proton exchangers that may play a role in maintaining the pH within exonemes.

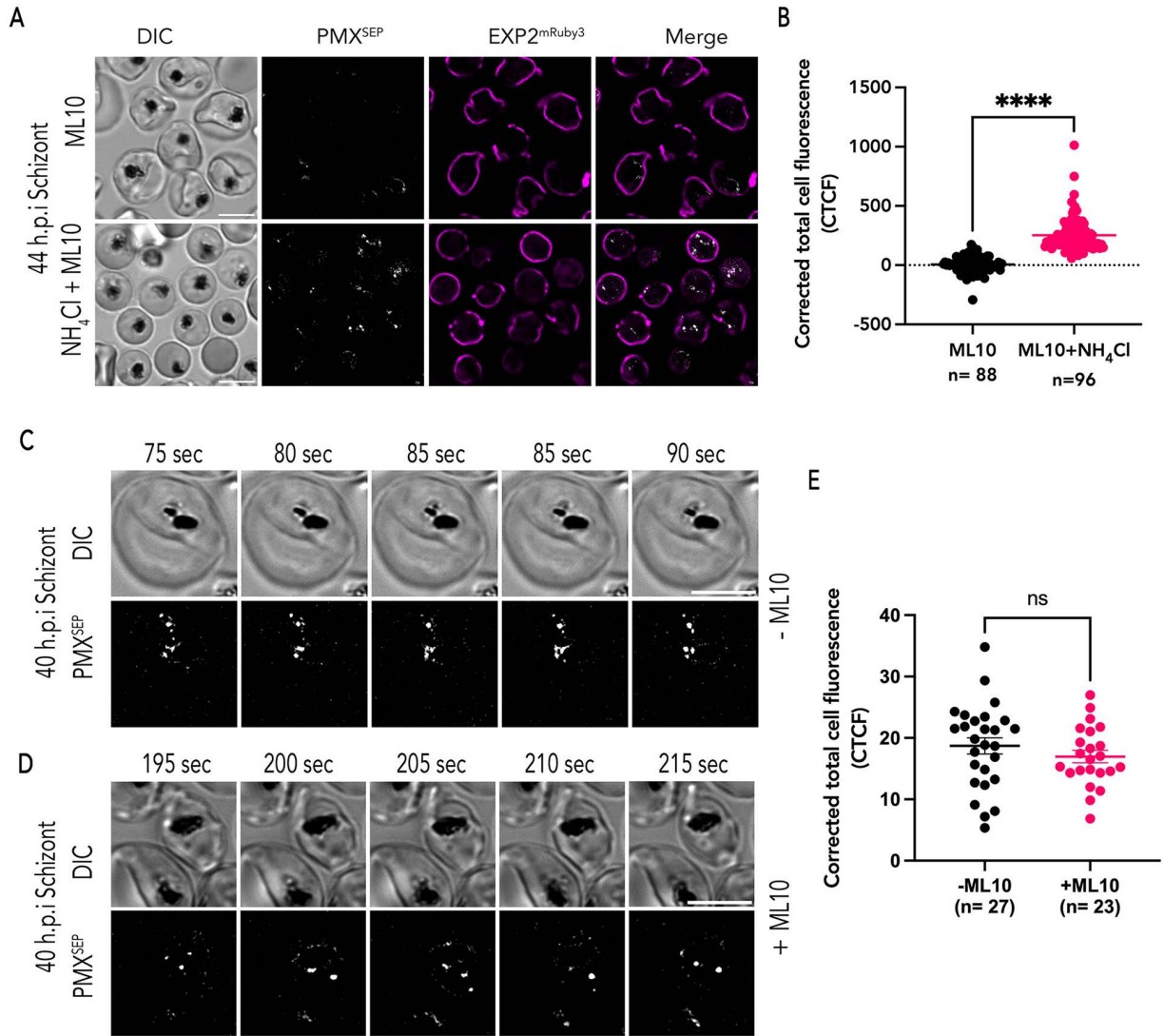

**Fig 6. A Protein kinase G inhibitor does not inhibit secretory vesicle acidification.** (A) Representative images from live imaging of synchronized 44 h.p.i PfEXP2$^{mRuby3}$/PMX$^{SEP}$ schizonts in the presence of ML10 (top) or ML10 as well as NH$_4$Cl (bottom). (B) Corrected total cell fluorescence (CTCF) of PMX$^{SEP}$ quantified from images of synchronized 44 h.p.i PfEXP2$^{mRuby3}$/PMX$^{SEP}$ schizonts in ML10 alone (black; n = 88, 3 biological replicates) or ML10 and NH$_4$Cl (magenta; n = 96, 3 biological replicates). ****p-value<0.0001, unpaired t-test. error bars = SEM. (C) Representative time lapse images of synchronized 40 h.p.i PfEXP2$^{mRuby3}$/PMX$^{SEP}$ acidification schizont. Snapshots of 5 consecutive time points are shown. (D) Representative time lapse images of synchronized 40 h.p.i PfEXP2$^{mRuby3}$/PMX$^{SEP}$ schizonts incubated with ML10. Snapshots of 5 consecutive time points are shown. (E) Corrected total cell fluorescence (CTCF) of PMX$^{SEP}$ quantified from images of synchronized 40 h.p.i PfEXP2$^{mRuby3}$/PMX$^{SEP}$ schizonts incubated without ML10 (black; n = 27, 3 biological replicates) or with ML10 (magenta; n = 23, 3 biological replicates). Unpaired t-test, ns = non-significant. error bars = SEM. Scale bar = 5 μm. Time-lapse recordings of acidification is a 5-sec interval.

## PfERC is required for exoneme exocytosis

Calcium signals have been suggested to be essential for exoneme exocytosis but the mediators of this calcium signal within the secretory pathway have not yet been identified [12,19]. We had previously identified that the *P. falciparum* endoplasmic reticulum-resident calcium-binding protein PfERC (PlasmoDB ID: PF3D7_1108600), EF-hand calcium-binding protein, is required for merozoite egress and regulates the maturation of PMX [32]. This protein belongs to a CREC family

of proteins (calumenin, reticulocalbin 1 and 3, ERC-55, and Cab) [33] and one member of this family is shown to interact directly with a protein to facilitate exocytosis [34]. However, the specific role of PfERC in exoneme egress remains unknown and we wanted to determine if PfERC functions in signal-dependent exocytosis of exonemes.

Our data showed that PfERC functions in the egress proteolytic cascade but is not required for ER calcium homeostasis or for microneme exocytosis [32]. Since egress proteases are stored within exonemes and released via exocytosis into the PV during egress, interfering with either exoneme biogenesis or exocytosis may result in loss of protease activity or function. Therefore, we hypothesized that PfERC is required for egress protease function because it may function in exoneme biogenesis or exocytosis.

To test this hypothesis, we first generated PfERC conditional mutants using the glucosamine (GlcN) responsive ribozyme system (Fig 7A) [32,35]. The addition of GlcN activates the *glmS* but not *M9* ribozyme, resulting in cleavage of mRNA that leads to mRNA degradation and protein knockdown (Fig 7B). The endogenous locus of PfERC in the NF54 *P. falciparum* parental strain was fused to three copies of the hemagglutinin (HA) tag and either the inducible *glmS* (termed PfERC-*glmS*) ribozyme or the inactive *M9* (termed PfERC-*M9*) ribozyme (Fig 7A) [32,35]. Clonal PfERC-*glmS* and PfERC-*M9* parasite lines were generated from independent transfections (S7A and S7B Fig). The PMX locus in these clonal PfERC-*glmS* and PfERC-*M9* parasite lines was then tagged with SEP to generate double mutants (termed PfERC-*glmS*/PMX$^{SEP}$ and PfERC-*M9*/PMX$^{SEP}$; S7C and S7D Fig). In these PfERC-*glmS*/PMX$^{SEP}$ and PfERC-*M9*/PMX$^{SEP}$ double mutants, PMX expression was also regulated by the *tetR* aptamer system [32,36] (S7 Fig).

Parasites were prepared as previously to obtain highly synchronized PfERC-*glmS*/PMX$^{SEP}$ and PfERC-*M9*/PMX$^{SEP}$ ring stage parasites (2 h.p.i). The 2 h.p.i ring stage parasites were incubated with 7.5 mM GlcN for 48 h. The addition of GlcN led to a reduction in PfERC expression in PfERC-*glmS*/PMX$^{SEP}$ schizonts, while there was no reduction in PfERC expression in PfERC-*M9*/PMX$^{SEP}$ schizonts grown under identical conditions (Fig 7B). These data confirmed that in these double mutants the addition of GlcN led to knockdown of PfERC only in PfERC-*glmS*/PMX$^{SEP}$ parasites (termed -PfERC) but had no effect on PfERC expression in PfERC-*M9*/PMX$^{SEP}$ parasites (termed +PfERC; Fig 7B). Our prior work had shown that PfERC knockdown led to a block in PMX activity [32,36]. The addition of GlcN to PfERC-*glmS*/PMX$^{SEP}$ and PfERC-*M9*/PMX$^{SEP}$ parasites also resulted in loss of PMX processing (Fig 7B).

Next, we wanted to test if PfERC knockdown had any effect on exoneme exocytosis. As before, we treated ring stage (2 h.p.i) PfERC-*glmS*/PMX$^{SEP}$ and PfERC-*M9*/PMX$^{SEP}$ parasites with GlcN and then these GlcN treated schizonts (44 h.p.i) were incubated with C1 for 4 hours to prevent egress, allow schizonts to fully develop and become egress competent. At 48 h.p.i, C1 was washed off from these schizonts and we observed these parasites using wide-field live fluorescence microscopy (Fig 7C). Time-lapse images were collected every 45 seconds for 30 minutes (Fig 7C). During the 30-min video, GlcN-treated PfERC-*MP*/PMX$^{SEP}$ schizonts were able to egress (Fig 7C and S10 Movie). Several fluorescent events were observed in PfERC-*M9*/PMX$^{SEP}$, while almost no fluorescence events were observed in GlcN-treated PfERC-*glmS*/PMX$^{SEP}$ (Fig 7C and S11 Movie). The corrected total cell fluorescence for each observed schizont within the field was integrated over 30 minutes (Fig 7D). Incubating PfERC-*M9*/PMX$^{SEP}$ with GlcN does not have an effect on exoneme exocytosis or egress (Fig 7D). In contrast, incubating PfERC-*glmS*/PMX$^{SEP}$ with GlcN results in loss of exoneme secretion and blocks egress (Fig 7D). It is suggested that PfERC is required for exoneme exocytosis (Fig 7C and 7D).

## Discussion

Merozoite egress from the infected RBC is a synchronized multistep process that is regulated in a timely manner [12]. Our current model for parasite egress suggests that two second messengers, calcium and cGMP, are critical for *Plasmodium* egress [12]. The molecular mechanisms linking calcium and cGMP are still unknown, but both these second messenger pathways are thought to converge on the exocytosis of egress-specific vesicles known as exonemes. Exoneme exocytosis is essential for parasite egress as it results in the release of two well-characterized proteases into the PV, SUB1 and PMX [10,13–16]. The release of SUB1 and PMX into the PV eventually results in PVM rupture, leading to parasite egress

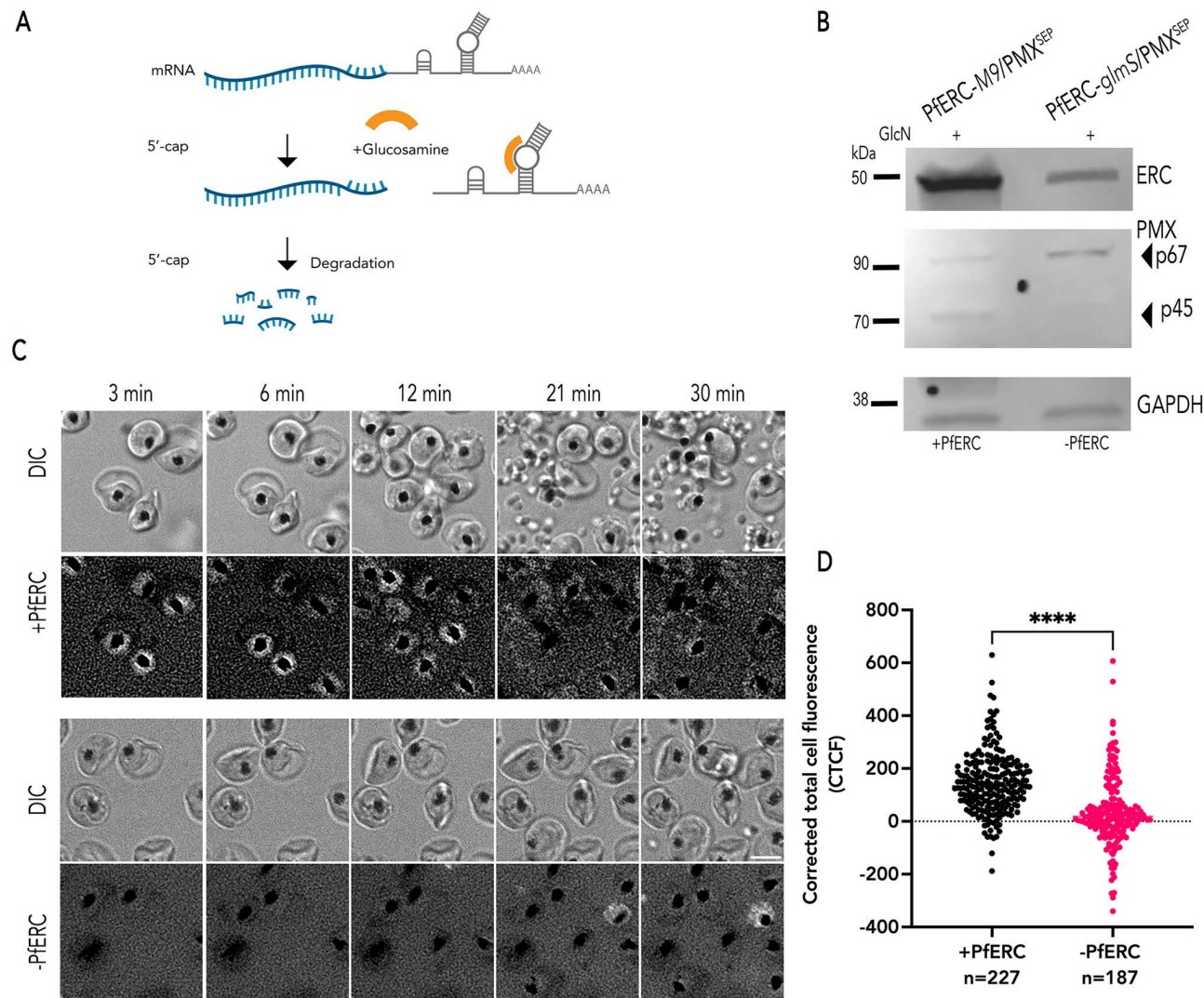

**Fig 7. PfERC is required for exoneme exocytosis in live parasites.** (A) Schematic shows the mechanism of the condition that knocks down the *glmS* ribozyme system. *glmS* is an inactive ribozyme that is transcribed but not translated with the mRNA of a protein of interest. Adding glucosamine (GlcN) leads to phosphorylation within the cell to glucosamine-6-phosphate (GlcN-6P). GlcN-6P binds to the transcribed mRNA, and the *glmS* ribozyme is activated and cleaves itself from the mRNA. This leads to disassociation of the mRNA from its poly(A) tail and results in the degradation of target-specific mRNA. The resulting decline in mRNA levels leads to reduced protein levels and, thus to loss of gene expression. As a control, we generated parasite lines containing a mutated version of the *glmS* ribozyme, called M9, which cannot cleave itself upon binding of GlcN. (B) Representative of western blot of parasite lysates isolated from PfERC-*glmS*/PMX^SEP and PfERC-*M9*/PMX^SEP schizonts, PfERC-*glmS* schizonts grown in the presence of GlcN for 48 h and probed with anti-HA antibodies (top panel), anti-GFP antibodies (middle panel), and anti-GAPDH (loading control; bottom panel). (C) Representative images from Glc-treated PfERC-*glmS*/PMX^SEP and PfERC-*M9*/PMX^SEP schizonts from live fluorescence microscopy. Phase contrast (top) and fluorescence (bottom) images. (D) Corrected total cell fluorescence (CTCF) integrated over the entire video (n = 10 recordings from 3 biological replicates for Glc-treated PfERC-*M9*/PMX^SEP, n = 11 recordings from 3 biological replicates for Glc-treated PfERC-*glmS*/PMX^SEP, ****p-value<0.0001, unpaired t-test). Scale bar = 5 µm.

[12]. However, the signal-dependent exocytosis of exonemes has not yet been directly observed in live *Plasmodium* schizonts during parasite egress. Thus, we aimed to determine the spatiotemporal relationship between the release of PMX into the PV and eventual rupture of the PVM. In this study, we used multiple approaches, including genetic tools combined

with live fluorescence microscopy, to investigate two critical steps during egress: exoneme exocytosis and PVM rupture in live parasites. Based on our results, we propose a timeline of *Plasmodium* egress from the infected RBC (Fig 8).

Studying exocytosis in live *Plasmodium* poses many challenges. Merozoites are only 1 µm in size, and nearly two dozen merozoites are packed in the PVM inside the ~7 µm diameter of the host RBC. Moreover, a group of merozoites are moving around the residual body within the schizont, which makes it extremely challenging to observe individual merozoites as well as to track the path of individual secretory organelles rapidly undergoing exocytosis. A reporter that provides a signal only upon exocytosis is ideal to monitor this process within *P. falciparum* schizonts. Therefore, in this study we used pH-sensitive variant of green fluorescent protein (GFP) known as superecliptic pHluorin or SEP [20,21] to monitor the exocytosis of exonemes during *Plasmodium* egress together with PVM morphology as well as determine if PfERC functions in this process. We targeted SEP to exonemes by fusing it with PMX, which did not interfere with PMX processing or function. Our data show that PMX$^{SEP}$ is non-fluorescent after acidification and is fluorescent upon exocytosis or neutralization with a weak base, with the two processes separated by a few hours during schizogony. In early schizogony (~40 h.p.i), we observed transient PMX$^{SEP}$ fluorescence that was insensitive to PKG inhibition, suggesting that these vesicles were undergoing acidification. Indeed, treating parasites with a weak base in later schizonts (~44 h.p.i) resulted in PMX$^{SEP}$ fluorescence showing that the protein is present in acidic vesicles. Since we observed vesicle acidification in early schizonts (~40 h.p.i) and vesicle exocytosis did not occur until later in schizogony (~44 h.p.i), it suggests that vesicle biogenesis and exocytosis are temporally separated.

Surprisingly, we did not observe loss of acidification when we used the V-type ATPase proton pump inhibitor BaF. A prior study localized the *P. falciparum* V-type ATPase to secretory vesicles in schizonts and showed that BaF leads to acidification within the parasite [31]. Since the proteases essential for egress are activated by low pH [15,37–40], this suggests that the V-type ATPase may not be responsible for acidification of exonemes. Proton pumping pyrophosphatases are another mechanism to acidify intracellular organelles and *P. falciparum* encodes for two pyrophosphatases, PfVP1 (PF3D7_1456800) and PfVP2 (PF3D7_1235200). PfVP1 localizes to the parasite plasma membrane and is required for ring to trophozoite stage transition [41] while PfVP2 is not essential for the intraerythrocytic life cycle [26,42]. Thus, it is not known how exonemes are acidified and more research is needed to identify the mechanism responsible for exoneme acidification.

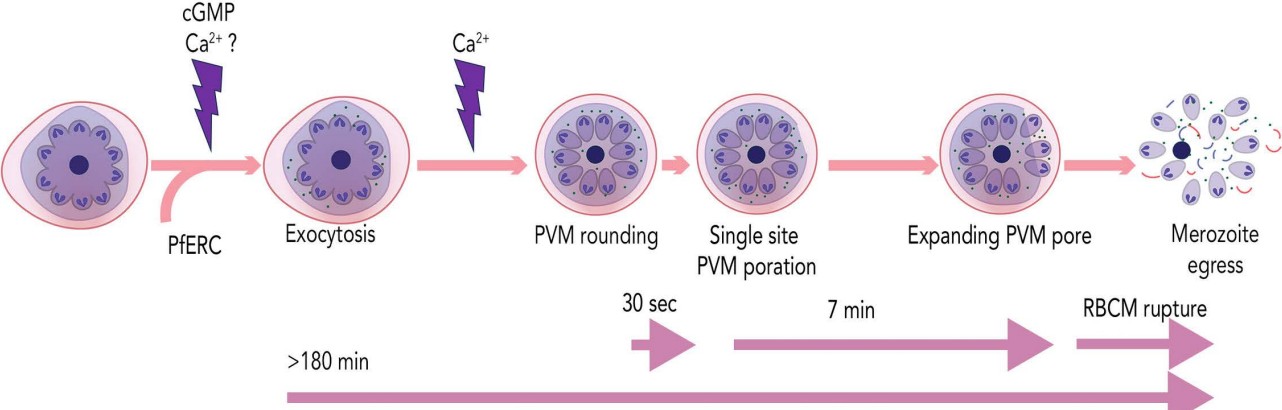

**Fig 8. A proposed model for *Plasmodium* egress.** Approximately 3 hours prior to egress, an unknown trigger leads to a rise in cGMP and may be calcium, resulting in the exocytosis of exonemes. This continues until merozoite egress. Exoneme exocytosis also requires PfERC. In 8-10 min prior to egress, a calcium signal triggers the PVM rounding, rapidly followed by its rupture at a single site. The PVM rupture rapidly expands for several minutes before the RBCM ruptures releasing merozoites from the infected RBC.

PMX[SEP] is non-fluorescent within the acidic exonemes unless we neutralized the exoneme pH using the weak base, ammonium chloride. We observe PMX[SEP] fluorescence prior to exoneme acidification in early schizonts (~40 h.p.i) or when it is exposed to the neutral environment within the PV upon signal-dependent exocytosis later during schizogony (~44 h.p.i). There was no reproducible change observed in the frequency or intensity of exocytosis during egress. Given that merozoites moved around the residual body, we could not track the path of each exocytosis event. Due to the resolution limits of microscopy, it was also not possible to locate the exact site of exoneme exocytosis or if it occurred at the apical end of merozoites. Nevertheless, our data show that blocking PKG signaling inhibits PMX[SEP] fluorescence but does not block exoneme acidification. Studies utilizing reversible inhibitors suggest that the cGMP and calcium signaling pathways that regulate merozoite egress are activated in the final minutes of the intraerythrocytic life cycle [2–11,19]. Our data suggest that exoneme exocytosis begins several hours prior to egress and continues until parasite egress. Since this process is dependent upon PKG signaling, these data further suggest that the cGMP signaling pathway may be activated when merozoites begin segmentation, several hours before segmentation is completed. These data are in agreement with studies showing that blocking merozoite segmentation does not inhibit parasite egress [43]. This is likely because, as our data shows, the egress pathway starts when segmentation begins and these processes are temporally linked but may occur independently or are linked earlier in schizogony before separating into independent pathways. It is also possible that the cGMP and calcium signals that kickstart exoneme exocytosis about 3 hours prior to egress are similar though not identical to the calcium signaling pathway that results in PVM rounding [2–11]. Alternatively, exoneme exocytosis requires a lower threshold of cytoplasmic calcium mobilization or only requires a cGMP signal and does not depend upon cytoplasmic calcium mobilization. Indeed, conditional knockout of the putative *P. falciparum* calcium channel, ICM1, did not prevent exocytosis of SUB1 into the PV suggesting that a calcium signal may not be required for exoneme exocytosis [44].

Even though our results suggest that the egress proteases are being released into the PV several hours prior to egress, it is not clear whether they are active in the PV during this time before egress. The egress proteases are activated by acidic pH [15,37–40] and it is possible that in the neutral pH of the PV, the proteases need several hours to ensure complete processing of their substrates in this sub-optimal environment. Another possibility is that they may have specific interacting partners that lead to an enhancement of protease activity [45]. More studies are needed to understand how the egress proteases function within the PV to finish the processing of their substrates to ensure the merozoites are invasion competent prior to egress.

Our findings are in agreement with prior studies that show that PVM rounding occurs about 8–10 minutes prior to egress [2–11,19]. A minute after PVM rounding, the PVM ruptures and our studies show that the timescale for rupture after membrane rounding was similar regardless of imaging timescale. The imaging data further show that the PVM ruptures with a < 3 µm pore at a single site on the membrane, which then rapidly expands. In all our biological replicates, we always observed only one site of PVM rupture. The current model for egress suggests that PVM rupture occurs because exonemes secrete proteases into the PV minutes prior to egress [8,10,13–18]. However, our data show that proteases are released into the PVM several hours prior to PVM rupture. Further, the mechanism responsible for PVM rupture remains unknown. Thus, it is not clear what prevents premature PVM rupture or how it is targeted only to the PVM and does not target the merozoite membrane or how this process is timed. There are several possible scenarios that may explain why PVM rupture does not occur until merozoite segmentation is finished. It is possible that the unknown membrane rupture protein or complex is not synthesized or released into the PV minutes prior to egress. Another possibility is that the protein responsible for PVM rupture is not activated until the PVM rounding occurs, leading to rapid membrane insertion and rupture. A recent study showed that a calcium signal precedes PVM rounding [11] and it is likely that the same signal activates the membrane rupture mechanism. Our data suggest that the protein (or complex) responsible for PVM rupture should either be able to form a ~3 µm pore that expands rapidly or this protein (or complex) destabilizes the PVM at a single site that rapidly propagates. Alternatively, a single protein or complex is not responsible for PVM rupture, and that

degradation of PV membrane proteins by egress proteases leads to membrane destabilization and rupture, though it is not clear why that would occur at a single site.

The step-by-step pathway that results in intracellular calcium release during egress remains to be determined. Our prior studies identified a key calcium-binding protein, PfERC, that was required for maturation of the egress protease, PMX [32]. However, a later study showed that blocking PMX maturation did not inhibit its proteolytic activity [37]. One possible model that would reconcile these two studies [32,37] would be that PfERC functions in exoneme exocytosis or biogenesis, which could result in inhibition of PMX maturation as well as secretion leading to a block in egress. Therefore, we sought to determine if PfERC functions in exoneme exocytosis or biogenesis. We show that knockdown of PfERC results in loss of exoneme exocytosis. These data suggest that PfERC functions upstream of the calcium signal responsible for PVM rounding. Since PfERC is primarily an ER-resident protein, PfERC could function in the biogenesis of exonemes and loss of PfERC leads to defects in biogenesis of exonemes resulting in inhibition of exocytosis. It is likely that defects in exoneme biogenesis may lead to loss of vesicle acidification. However, we did not observe any loss in acidification of exonemes upon PfERC knockdown, these data suggest that any function PfERC may have in exoneme biogenesis does not neutralize exoneme pH. It is also likely that PfERC may function in an earlier cGMP and/or calcium signal required for exoneme exocytosis. It remains to be determined whether the calcium binding function of PfERC is required for its role in exoneme exocytosis.

These studies demonstrate the development of a live-cell reporter to observe signal-dependent exocytosis required for egress of *P. falciparum* from the infected RBC. Using this reporter, we discovered that signal-dependent exoneme exocytosis starts several hours prior to the egress of merozoites. These data suggest that there may be two independent signals for egress, one for starting exoneme exocytosis several hours prior to egress and another that results in PVM rounding as well as rupture a few minutes before egress (Fig 8). The data identified a calcium binding protein, PfERC, as a critical player in exoneme exocytosis that may function in the calcium signaling pathway required for exocytosis. These data show that PKG signaling is critical for exoneme exocytosis and occurs several hours prior to egress. It is not yet clear whether this results in a continuous calcium signal or if that occurs later during egress. This study lays the groundwork for dissecting the molecular mechanisms utilized by the deadly malaria-causing parasites to egress from their host cells, which are likely to be similar across the parasitic lifecycle of *P. falciparum.*

## Materials and methods

### Plasmid construction

Genomic DNA was isolated from NF54[attb] *P. falciparum* cultures using a QIAamp DNA Blood Kit (Qiagen), and all constructs were confirmed by sequencing. Briefly, PCR products were inserted into the respective plasmids using the sequence- and ligation-independent cloning (SLIC) method. A mixture of an insert, a cut vector, and T4 DNA polymerase was incubated for 2.5 minutes at room temperature and then for 10 minutes on ice. A bacterial transformation uses a heat shock at 42 °C for 35 sec. All primers utilized in this study are in S1 Table. Two plasmids, pbPM2-EXP2-mRuby [26] and pAIO2-EXP2 gRNA [46], were gifts from Dr. Josh Beck, Iowa State University.

SEP was amplified from pHmK-C1 (Addgene) [25] using primers P10/P11. The pPMX-V5-Apt-pMG74 plasmid [32] was modified to replace V5 tag with THE SEP amplicon between PsPXi and AflII cut site, resulting in a pPMX-SEP-Apt-pMG74 plasmid. A PMX guide RNA targeting the 30 bp upstream *pmx* stop codon was annealed using P12/P13. The annealed fragment was inserted into the plasmid pUF1-Cas9, which contains the DHOD resistance gene, resulting in the plasmid pUF1-Cas9-PMX.

Primers P14/15 were used to amplify an 843 bp fragment of 5′ homology PfERC C-terminus (excluding the stop codon) from *P. falciparum* NF54[attb] genomic DNA. Amplicons were inserted into pHA-SDEL-glmS and pHA-SDEL-M9 using restriction sites SacII and AfeI [32]. A 831-bp 3′ homology PfERC 3UTR (beginning at 13 bp downstream PfERC stop codon) was amplified by primers P16/P8 and was inserted into pHA-SDEL-glmS and pHA-SDEL-M9 containing 5 ′ homology PfERC C-terminal region at restriction sites HindIII and NheI, resulting in pPfERC-HA-SDEL-glmS or

pPfERC-HA-SDEL-M9 [32]. A gRNA targeted at 27 bp upstream of the *erc* stop codon. The PfERC gRNA was annealed using P17/P18 and inserted into the plasmid pUF1-Cas9-guide, resulting in the plasmid pUF1-Cas9-ERC.

## Parasite culturing and transfection

All transfections were done in duplicate. For transfections, an uninfected RBC pellet was electroporated with a matching pair of 20 µg of donor plasmid and 50 µg of CRISPR/Cas9 gRNA plasmid using a Bio-Rad Gene Pulser Xcell Electroporation System. The parasite line was then added to the transfection mixture and cultured in RPMI medium supplemented with Albumax I (Gibco) and the required drugs described previously [32,47–50]. All primers used are in S1 Table.

To generate PfEXP2mRuby3 parasites, 20 µg of the pbPM2-EXP2-mRuby donor plasmid was linearized with AflII overnight, and 50 µg of the pAIO-EXP2-gRNA were co-transfected into NF54attb parasites [51]. At 24 h after transfection, 2.5 µg/ml of Blasticidin-S (BSD) was added to select for pbPM2-EXP2-mRuby expression [26,42]. The parasite population was cloned by limiting dilution. A PCR diagnostic for integration was performed using primers P1/P3 to amplify a 2.1 kb fragment corresponding to the correct integrant.

For generating PfEXP2mRuby3/PMXSEP parasites, 20 µg of pPMX-SEP-Apt-pMG74, a donor plasmid, was linearized with EcoRV overnight, and 50 µg of pUF1-Cas9-PMX gRNA plasmid were co-transfected into PfEXP2mRuby3 parasites [13,32]. Transfected parasites were grown in 0.5 µM anhydrous tetracycline (aTc) (Cayman Chemical) and 2.5 µg/ml of Blasticidin-S (BSD) [32,42]. At 48 h post-transfection, Cas9 expression was selected by drug cycling [52,53] in the presence and absence of 1 µM DSM1 for 4 days until parasites were detected in the culture. Next, the parasite population was cloned by limiting dilution. A PCR diagnostic for integration was performed using primers P4/P6 to amplify a 2.8 kb fragment corresponding to the correct integrant.

For generating the PfERC-glmS and PfERC-M9 parasites, a mix of two plasmids (20 µg of a pPfERC-HA-SDEL-glmS or pPfERC-HA-SDEL-M9 donor plasmid and 50 µg of pUF1-Cas9-ERC guide plasmid) was co-transfected into NF54attb parasites. To select for Cas9 expression, 1 µM DSM1 was applied to the transfected culture as previously described [32]. Clonal parasites were subjected to PCR diagnosis using primers P7/P8 and P7/P9 to amplify a 2.2-kb and 1.1-kb fragment, respectively, for correct integration [32].

To generate PfERC-glmS/PMXSEP and PfERC-M9/PMXSEP parasites, 20 µg of pPMX-SEP-Apt-pMG74, a donor plasmid, was linearized with EcoRV overnight, and a 50 µg of pUF1-Cas9-PMX gRNA plasmid were co-transfected into PfERC-glmS or PfERC-M9 parasites [13,32]. Transfected cultures were grown in 2.5 µg/ml of blasticidin S (BSD) [32,42]. To select a donor plasmid, 0.5 µM anhydrous tetracycline (aTc) (Cayman Chemical) was applied at 48 h post-transfection. A PCR diagnostic for integration was performed using primers P4/P6 to amplify a 2.8 kb fragment corresponding to the correct integrant.

## Growth assay

Asynchronous parasites were placed in 6 well-plates at 0.3% parasitemia at 2% hematocrit in triplicate and grown for 6 days. For growth assessment, samples were taken three technical replicates every 48 hours. For each technical replicate, 5 µL of culture from each well was resuspended in 200 µl of 8 µM Hoechst (ThermoFisher Scientific) in PBS. Parasites were incubated shaking in the solution for 20 min at room temperature before being assessed via flow cytometry on The Quanteon (Agilent) analyzer. Flow cytometry analysis was done using FlowJo (Tree Star, Inc) and data were graphed using GraphPad Prism (Prism 10 for macOS Version 10.2.2 (341).

## Western blotting

Western blotting was performed using *P. falciparum* lysates as described previously [32] with minor modifications. Highly synchronized late-schizont parasite culture pellets were treated with ice-cold 0.05% saponin–phosphate-buffered saline (PBS) for 15 min, followed by sonication. The sonicated lysates were collected for detection of proteins separated via a western blot assay. The antibodies used in this study were rabbit anti-HA (715500; Invitrogen) (1:1000), mouse anti-GFP

(JL-8; Takara) (1:3000), rabbit anti-EXP2 (from H.Ke) (1:10000), and mouse anti-GAPDH (1.4; Millipore Sigma)(1:2000). The secondary antibodies used were IRDye 680CW goat anti-rabbit IgG and IRDye 800CW goat anti-mouse IgG (Li-COR Biosciences) (1:20,000). The Western blotting images were processed using Odyssey Clx Li-COR infrared imaging system software (Li-COR Biosciences).

### Time-lapse imaging of live *P. falciparum*

To prepare highly synchronized parasites, late schizont parasites were incubated with C1 or ML10 to prevent egress, then washed off the compound. After allowing parasite egress and invasion for 2 hours, we perform 5% sorbitol on the culture to collect 0–2 h.p.i rings. The rings were grown in media supplemented with mixed gas in a 37 °C shaker incubator.

Late-stage schizont parasites (44 h.p.i) were isolated on a Percoll gradient (Genesee Scientific). Percoll pellet was incubated in the PKG inhibitors either 1.5 µM compound 1 (C1) {4-[2-(4-fluorophenyl)-5-(1-methylpiperidine-4-yl)-1H-pyrrol-3-yl] pyridine} [8,18] or 25 nM ML10 (BEI Resources, catalog no. NR-56525, www.beiresources.org) [54] at 37 °C in a $CO_2$ incubator. After 4 h incubation, the egress inhibitor was washed off twice using pre-warmed 37°C cRPMI media. After washing off the egress inhibitor, 200–500 µL schizont-stage parasite culture was spun down at 3000 rpm for 30-sec, and the supernatant was removed. The pellet was gently resuspended with warm (37°C) RMPI media, and the tube was spun down at 3000 rpm for 1 min. After removing the supernatant, the pellet was gently resuspended with 200–500 µL warm (37°C) media before being placed onto a 35 mm glass-bottom dish coated with poly-L-lysine for imaging.

The 30-minute time-lapse video was performed using a DeltaVision II microscope with autofocus. The PfEXP2[mRuby3]/PMX[SEP] schizonts were imaged with a Live cell filter set CFP, YFP, GFP, and mCherry. To minimize photodamage, the laser power level was kept at 10% (or less). A 488-nm-wavelength laser was used to excite PMX[SEP], and a 561-nm-wavelength laser was used to excite PfEXP2[mRuby3]. Time-lapse images were taken every 30 sec up to 30 min at 37 °C in an imaging chamber supplemented with $CO_2$.

The 5-h time-lapse video was imaged using a Zeiss LSM 980 Confocal system using an inverted Axio Observer 7 microscope stand with transmitted (HAL), UV (HBO), and laser (405 – 730 nm) illumination sources. A 488-nm-wavelength laser (10% power) was used to excite PMX[SEP], and a 561-nm-wavelength laser (2% power) was used to excite PfEXP2[mRuby3]. The microscope system uses Zen 3.3 (Blue) acquisition software with autofocus. The dark incubation chamber with regulated $CO_2$ (5%), temperature (37 °C), and humidity was used. Images were captured every 5 minutes for 5 hours.

To treat parasites with BaF, highly synchronized PfEXP2[mRuby3]/PMX[SEP] parasites were prepared as described. The 44 h.p.i schizont pellet was incubated with ML10 for 2–4 h. Before imaging, 100 nM BaF (Cayman Chemical) was added to the culture and incubated for 10 min.

### Imaging acidification of exonemes

The 40 h.p.i PfEXP2[mRuby3]/PMX[SEP] schizonts were separated using the Percoll gradient (Genesee Scientific). The schizont pellet was either incubated in media supplemented with and without 25 nM ML10 (BEI Resources, catalog no. NR-56525, www.beiresources.org) [54] at 37 °C in a $CO_2$ incubator. For imaging, 200–500 µl of ML10-schizont culture was placed onto a 35 mm glass-bottom dish coated with poly-L-lysine. The 15-minute time-lapse video was performed using a DeltaVision II microscope with autofocus. The 40 h.p.i PfEXP2[mRuby3]/PMX[SEP] schizonts were imaged with a Live cell filter set CFP, YFP, GFP, and mCherry. The laser power level was kept at 32% (or less). A 488-nm-wavelength laser was used to excite PMX[SEP]. Time-lapse images were taken every 5 sec up to 15 min at 37 °C in an imaging chamber supplemented with $CO_2$.

### Ammonium chloride treatment of live *P. falciparum*

Ammonium chloride treatment was carried out at two time points: 44 h.p.i and 48 h.p.i PfEXP2[mRuby3]/PMX[SEP] schizont parasites. For both timepoints, schizonts were separated using the Percoll gradient (Genesee Scientific). The schizont pellet

was incubated media supplemented with 25 nM ML10 (BEI Resources, catalog no. NR-56525, www.beiresources.org) [54] at 37 °C in a CO2 incubator. For the control, 200–500 μL ML10-schizont culture was placed onto a 35 mm glass-bottom dish coated with poly-L-lysine for imaging. For ammonium chloride treatment, 200–500 μL ML10-schizont culture was resuspended with ammonium chloride at a final concentration of 20 mM in ML10 supplemented medium. Ammonium chloride was incubated for 5–10 minutes before imaging.

### Image processing and analysis

ImageJ (Fiji version 2.0.0-rc-68/1.52h) was used to crop images, adjust brightness and intensity, overlay channels, and prepare montages. The adjustments to brightness and contrast were made only for display purposes. The fluorescence intensity of each cell was calculated using the calculation for corrected total cell fluorescence (CTCF) = integrated density–(area of selected cell × mean fluorescence of background readings) [55,56]. The data for calculation were obtained only from the SEP fluorescence channel as follows: 1) From the *Analyze* menu on ImageJ, *set measurements* were selected with the area, min, max gray values, and mean gray values. 2) The area of the cell of interest was manually selected using a freehand tool. 3) The selected area was measured using *Measure*. 4) Step 2–3 was repeated for every time point. 5) For background readings, an outline was drawn on the space outside the cell of interest. 6) The measure menu was used to measure fluorescence as described above. 7) The fluorescence intensity of each cell was calculated using Excel (Microsoft Excel for Mac Version 16.103 (25110922)). Plots and statistical analysis (2-sided unpaired Student t-tests) were done using GraphPad Prism (Prism 10 for macOS Version 10.2.2 (341).

### Supporting information

**S1 Fig. Replication curve of PfEXP2<sup>mRuby3</sup> and PfEXP2<sup>mRuby3</sup>/PMX<sup>SEP</sup> parasites.** Parasitemia of PfEXP2<sup>mRuby3</sup> (black) and PfEXP2<sup>mRuby3</sup>/PMX<sup>SEP</sup> (magenta) is measured via flow cytometry. Representative of 3 biological replicates. Each data point represents the mean of three technical repeats. (error bars = SD; not significant by unpaired t-test).
(JPG)

**S2 Fig. ML10 PKG inhibition blocks exoneme exocytosis.** (A) A representative image from live imaging of synchronized PfEXP2<sup>mRuby3</sup>/PMX<sup>SEP</sup> schizonts in the presence of ML10 (B) A representative image from live imaging of synchronized PfEXP2<sup>mRuby3</sup>/PMX<sup>SEP</sup> schizonts after ML10 removal. Parasite egress occurs, and free-merozoites are scattered in the extracellular space (DIC). (C) Corrected total cell fluorescence (CTCF) of PMX<sup>SEP</sup> quantified from time-lapse images of synchronized PfEXP2<sup>mRuby3</sup>/PMX<sup>SEP</sup> schizonts incubated with ML 10 (black; n = 44, 2 biological replicates) or without ML10 (magenta; n = 33, 2 biological replicates) C1. ***p-value<0.0001, unpaired t-test. error bars = SEM Scale bar = 5 μm.
(JPG)

**S3 Fig. Measuring PVM diameter and split ends.** A representative of measuring PVM diameter (A-C). By drawing a line across the rounded PVM (A), measuring fluorescence intensity values corresponding to the lines are shown in (B), and generating a normalized fluorescence intensity graph in (C), the fluorescence intensity peaks are labelled as A and A'. A representative of measuring the distance between two split ends of PVM (D-F). Drawing a line across two split ends labeled as Peak B and B' (C) and measuring fluorescence intensity values corresponding to the line (D) and its normalized fluorescence intensity graph generated (F).
(JPG)

**S4 Fig. Exonemes are acidified prior to exocytosis.** (A-B) Constant PMX<sup>SEP</sup> were detected at the 0 min till PfEXP2<sup>mRuby3</sup>/PMX<sup>SEP</sup> schizonts egress. Time-lapse images were taken at 5-min intervals. Scale bar = 5 μm; n = 1 biological replicates. (C-D) Representative images of live cell imaging of PfEXP2<sup>mRuby3</sup>/PMX<sup>SEP</sup> schizonts in the presence of ML10.

All schizonts show no fluorescence detected throughout the 5-hour recording. Time-lapse images were taken at 5-min intervals. Scale bar = 5 μm; n = 2 biological replicates.
(JPG)

**S5 Fig. Normalized PMX**<sup>SEP</sup>**intensity of 13 schizonts.** Corrected total cell fluorescence (CTCF) values were used to normalize against the highest CTCF of each individual schizont. Colors represent each individual schizont.
(JPG)

**S6 Fig. Exoneme acidification is insensitive to V-type ATPases-specific inhibitor.** (A) A representative still image from a time-lapse video of ML10-treated PfEXP2$^{mRuby3}$/PMX$^{SEP}$ late schizont and (B) A representative still image from a time-lapse video of BaF-ML10-treated PfEXP2$^{mRuby3}$/PMX$^{SEP}$ late schizont. (C) Corrected total cell fluorescence (CTCF) values of PMX$^{SEP}$ schizonts. Black dot represents the CTCF value of each schizont with ML10 (n = 63, 3 replicates). Pink dot represents the CTCF value of each schizont in the presence of ML10 and BaF(n = 63, 3 replicates), ns = not significant, unpaired t-test Scale bar = 5 μm.
(JPG)

**S7 Fig. Generation of PfERC conditional mutants expressing PMX**<sup>SEP</sup>**.** (A) Schematics of the targeting (pERC-TOPO) plasmids by the CRISPR/Cas9 system and a guide RNA. The locations of diagnostic primers (P7, P9, and P9) used to demonstrate the repair of the locus via double-crossover homologous integration are shown. (B) Agarose gel showing the PCR diagnostic test using the primer pairs P7 + P8 (left) and P7 + P9 (right). Amplicons were amplified from the genomic DNA extracted from parental and mutant parasites. (C) Schematics of the targeting *pmx* with SEP. (D) PCR diagnosis of SEP into the PfPMX locus in NF54$^{attb}$, PfERC-*M9*, PfERC-*M9*/PMX$^{SEP}$, PfERC-*glmS* and PfERC-*glmS*/PMX$^{SEP}$ using primer pair (P4 and P6).
(JPG)

**S1 Table. List of primers used in the study to generate the parasite lines.**
(DOCX)

**S1 Movie. Time-lapse imaging of PfEXP2**$^{mRuby3}$**/PMX**<sup>SEP</sup> **schizonts in the presence of C1.** Representatives of PfEXP2$^{mRuby3}$/PMX$^{SEP}$ parasites within RBCs until the end of recording (DIC, left). PfEXP2$^{mRuby3}$ signals were irregularly shaped, and PMX$^{SEP}$ was not detected (Merged two fluorescence channels, right) (5 recordings from 3 biological replicates). Scale bar = 5 μm.
(M4V)

**S2 Movie. Time-lapse imaging of PfEXP2**$^{mRuby3}$**/PMX**<sup>SEP</sup> **schizonts in the absence of C1.** Representative of PfEXP2$^{mRuby3}$/PMX$^{SEP}$ parasites egress upon C1 removal, and free merozoites into the extracellular space (left). PMX$^{SEP}$ fluorescence was detected from the beginning of the recording until after merozoite egress occured. PfEXP2$^{mRuby3}$ signals were irregularly shaped (right). n = 6 recordings from 3 biological replicates. Scale bar = 5 μm.
(M4V)

**S3 Movie. Time-lapse imaging of exocytosis and egress of PfEXP2**$^{mRuby3}$**/PMX**<sup>SEP</sup> **schizonts.** Late schizont PfEXP2$^{mRuby3}$/PMX$^{SEP}$ parasites (left). Exoneme exocytosis of PfEXP2$^{mRuby3}$/PMX$^{SEP}$ was detected as punctate patterns (Middle). PVM morphology changed from irregular shape to a rounded shape before PVM breakdown (right), (7 biological replicates, double-labeled parasites, time-lapse recordings of parasite egress are at a 30-sec interval). Scale bar = 5 μm.
(M4V)

**S4 Movie. 4 sec- interval of time-lapse imaging of PfEXP2**$^{mRuby3}$**/PMX**<sup>SEP</sup> **schizonts shows PVM rupture at a single site.** Representative of PfEXP2$^{mRuby3}$/PMX$^{SEP}$ late schizont (left). PfEXP2$^{mRuby3}$ signals were irregularly shaped and transitioned to rounded before rupture at a single site (right). Scale bar = 5 μm (5 recordings from 2 biological replicates).
(M4V)

**S5 Movie. 2 sec- interval of time-lapse imaging of PfEXP2$^{mRuby3}$/PMX$^{SEP}$ schizonts showed the PVM rupture at a single site.** Representative of PfEXP2$^{mRuby3}$/PMX$^{SEP}$ schizonts showing PVM morphology changed from irregular to rounded before its breakdown at a single site. Scale bar = 5 μm (6 recordings from 4 biological replicates). (M4V)

**S6 Movie. Exocytosis of PfEXP2$^{mRuby3}$/PMX$^{SEP}$ schizonts was detected over 3 hrs before egress.** Punctate PMX$^{SEP}$ exocytosis signal was detected about 3 hours prior to egress (middle). PfEXP2$^{mRuby3}$ signals showed PVM morphology (right). PfEXP2$^{mRuby3}$/PMX$^{SEP}$ schizont egressed freeing merozoites into the extracellular space (left). Scale bar = 5 μm. (M4V)

**S7 Movie. Exocytosis of PfEXP2$^{mRuby3}$/PMX$^{SEP}$ schizonts is inhibited.** None of PMX$^{SEP}$ exocytosis signal was detected in the presence of ML10 throughout 5 hours of recording (right). PfEXP2$^{mRuby3}$/PMX$^{SEP}$ schizont could not egres (left). Time-lapse images were taken at 5-min intervals. Scale bar = 5 μm; n = 2 biological replicates. (M4V)

**S8 Movie. Acidification of PfEXP2$^{mRuby3}$/PMX$^{SEP}$ schizonts is detected.** PMX$^{SEP}$ acidification signal was detected during 40 h.p.i schizont. Time-lapse images were taken at 5-sec intervals over 15 minutes. Scale bar = 5 μm (10 recordings from 3 biological replicates). (M4V)

**S9 Movie. Acidification of PfEXP2$^{mRuby3}$/PMX$^{SEP}$ schizonts is not inhibited by protein kinase G ML10 inhibitor.** PMX$^{SEP}$ acidification signal was detected during 40 h.p.i schizont in the presence of ML10. Time-lapse images were taken at 5-sec intervals over 15 minutes. Scale bar = 5 μm (9 recordings from 3 biological replicates). (M4V)

**S10 Movie. Time-lapse imaging of GlcN-treated PfERC-_M9_/PMX$^{SEP}$ schizonts egress.** Representatives of GlcN-treated PfERC-_M9_/PMX$^{SEP}$ schizonts egress upon C1 removal (left). PMX$^{SEP}$ exocytosis signal is detected in schizonts (right). (10 recordings, n = 3 biological replicates). Scale bar = 5 μm. (M4V)

**S11 Movie. Glc-treated PfERC-_glmS_/PMX$^{SEP}$ schizonts block exocytosis and egress.** Representatives of GlcN-treated PfERC-_glmS_/PMX$^{SEP}$ schizonts egress upon C1 removal reside within the red blood cells (left). Almost all GlcN-treated PfERC-_glmS_/PMX$^{SEP}$ schizonts lose PMX$^{SEP}$ exocytosis signals (right). 11 recordings, n = 3 biological replicates. Scale bar = 5 μm. (M4V)

## Acknowledgments

We thank members of the Muralidharan lab and Matthias Garten for their thoughtful comments and suggestions; Josh Beck for the EXP2 tagging plasmids; Purnima Bhanot for compound 1; Muthugapatti Kandasamy at the University of Georgia (UGA) Biomedical Microscopy Core.

## Author contributions

**Conceptualization:** Watcharatip Dedkhad, Manuel A Fierro, Vasant Muralidharan.

**Data curation:** Watcharatip Dedkhad, Carrie Brooks.

**Formal analysis:** Watcharatip Dedkhad, Vasant Muralidharan.

**Funding acquisition:** Vasant Muralidharan.

**Investigation:** Watcharatip Dedkhad, Tia Tran, Manuel A Fierro, Carrie Brooks.

**Methodology:** Watcharatip Dedkhad, Manuel A Fierro, Carrie Brooks, Vasant Muralidharan.

**Project administration:** Vasant Muralidharan.

**Supervision:** Watcharatip Dedkhad, Carrie Brooks, Vasant Muralidharan.

**Validation:** Tia Tran, Carrie Brooks.

**Visualization:** Watcharatip Dedkhad, Vasant Muralidharan.

**Writing – original draft:** Watcharatip Dedkhad, Vasant Muralidharan.

**Writing – review & editing:** Watcharatip Dedkhad, Vasant Muralidharan.

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
