## [Decision Letter · Decision Letter 0]

7 Oct 2025

PPATHOGENS-D-25-02207

Spatiotemporal dynamics of signal dependent exocytosis and parasitophorous vacuolar membrane rupture during Plasmodium falciparum egress.

PLOS Pathogens

Dear Dr. Muralidharan,

Thank you for submitting your manuscript to PLOS Pathogens. After careful consideration, we feel that it has merit but does not fully meet PLOS Pathogens's publication criteria as it currently stands. Therefore, we invite you to submit a revised version of the manuscript that addresses the points raised during the review process.

Please submit your revised manuscript within 60 days Dec 06 2025 11:59PM. If you will need more time than this to complete your revisions, please reply to this message or contact the journal office at plospathogens@plos.org. Please include the following items when submitting your revised manuscript:

We look forward to receiving your revised manuscript.

Kind regards,

Mathieu Brochet

Academic Editor

PLOS Pathogens

Dominique Soldati-Favre

Section Editor

PLOS Pathogens

  Sumita Bhaduri-McIntosh

Editor-in-Chief

PLOS Pathogens

orcid.org/0000-0003-2946-9497

Michael Malim

Editor-in-Chief

PLOS Pathogens

orcid.org/0000-0002-7699-2064

**Additional Editor Comments (if provided):**

The three reviewers agree that the manuscript has strong potential to advance our understanding of exoneme exocytosis. However, they raise several important concerns that require experimental validation. This includes the reliance on live-cell imaging without complementary biochemical data to support the central claims. They also recommend further validation of the reporter assay, noting possible confounding effects related to vesicle formation, acidification, and exocytosis. Additionally, concerns were raised about potential off-target effects of the PKG inhibitors used, which could complicate interpretation of the results.

**Journal Requirements:**

**Reviewers' Comments:**

Reviewer's Responses to Questions

**Part I - Summary**

Reviewer #1: Dedkhad et al. study P. falciparum exoneme exocytosis and PVM rupture, events that precede merozoite egress and invasion of new erythrocytes. The studies involve sophisticated imaging of several edited parasite lines, conditional knockdown of a key regulator enzyme, and relevant pharmacological reagents (C1, ML10, and BaF). The authors’ transfectant lines are well-designed and have the potential to provide important insights into timing of key events (esp. discharge of exonemes, PVM ultrastructural changes and rupture, subsequent egress). Although the paper has excellent potential to advance our understanding of this critical, druggable stage in the malaria parasite life cycle, a number of issues require experimental controls and more thorough examination.

Reviewer #2: Dedkhad et al, describe the spatiotemporal dynamics of exoneme exocytosis and rupture of the PVM in relation to P. falciparum egress. The authors utilise a reporter (SEP) to visualise exoneme exocytosis in live P. falciparum schizonts by fusing it to Plasmepsin X, in a line where PVM breakdown can be in parallel monitored as EXP2, a known PVM marker, is fused to mRuby3. By using these tools, the authors confirm that exoneme exocytocis is PKG-dependent, as has been previously shown by other groups, and that PVM rupture occurs probably in a single site followed by full schizont egress within an average of 7 minutes. By using the SEP reporter the authors conclude that exoneme exocytosis occurs several hours prior to merozoite egress. Furthermore by using conditional genetics, they show that an ER-resident calcium binding protein (PfERC), which has been previously shown to be important for egress, is involved in exoneme exocytosis but not in the biogenesis of these organelles.

Overall, this study provides new insight on the complex proteolytic and signalling cascade that leads to schizont egress in Plasmodium, however certain conclusions seem to be premature and key aspects need to be addressed before it is suitable for publication.

Reviewer #3: The authors report a dual-reporter system to study the exocytosis of exonemes into the parasitophorous vacuole (PV) immediately prior to _Plasmodium_ egress. Egress is a rapid, multistep process that results in the rupture of the two bounding membranes: the PV membrane (PVM) and the red blood cell membrane (RBCM). Exonemal secretion is essential for this process, as it delivers key effectors, most importantly the proteases PMX and SUB1, to the PV where they initiate a proteolytic cascade leading to egress.

While previous studies have assessed these events and the localisation of the molecules involved in them primarily by IFAs and electron microscopy, this reporter system enables live imaging of exocytosis. The authors achieve this by tagging PMX with the pH-sensitive fluorophore SEP, which fluoresces only upon exposure to neutral pH following exocytosis, and the PVM-bound protein EXP2 with mRuby3 to continuously visualise the PVM. This approach provides, for the first time, a live view of exocytosis and PVM dynamics just prior to egress.

Using this system, they show that exocytosis can begin as early as three hours prior to egress, confirm characteristic PVM dynamics like rounding consistent with prior studies, and suggest that “the PVM ruptures at a single site.” The authors convincingly validate their system by demonstrating the absence of SEP signal when egress is stalled by PKG inhibition (which prevents initiation of calcium signalling) or in an ERC knockdown line (ERC being required for egress-specific vesicle formation).

Overall, I found the majority of the experiments to be robust, well-presented, and clearly described. The paper is well-written and easy to follow, and I recognise the considerable effort involved in live imaging such a challenging process. However, in some instances, the authors’ interpretation seems to extend beyond the evidence presented, and additional controls or experiments would strengthen the conclusions.

**Part II – Major Issues: Key Experiments Required for Acceptance**

Reviewer #1: 1. The authors’ proposal that exoneme exocytosis may be triggered as early as 3 h prior to merozoite egress contrasts with the prevailing view (model proposed in PMID: 28683142), where exoneme discharge leads to SERA5 processing and egress within 8 min. I understand that the authors used live-cell imaging that might capture exocytic events missed by prior studies. However, a few considerations suggest more cautious interpretation is warranted.

a. The schematic in Fig. 1A shows the PMXSEP reporter initially producing a fluorescent signal upon packaging into nascent non-acidified exonemes; subsequent acidification causes loss of fluorescence. Fusion with the PPM neutralizes the fused vesicle to uncover SEP fluorescence as a transient punctum that dissipates due to dilution in the vacuolar space. As the kinetics of exoneme formation and acidification are unknown, it seems likely that PMXSEP packaging and subsequent exoneme acidification will produce a similar transient punctum. Moreover, because PMX transcription peaks around 40-42 h (per DeRisi lab microarray data on PlasmoDB), this reviewer expects to see fluorescence puncta associated with packaging.

b. Comparison of Fig. 2A and 2D shows that Compound C1 completely blocks invasion (Fig. 2A) but only partially reduces cellular fluorescence (Fig. 2D). Given the critical role of PKG in exoneme discharge, this is very surprising. To this reviewer, ongoing fluorescence speckles in the presence of C1 appears to reflect synthesis and packaging of PMXsep into new exonemes and subsequent loss of fluorescence due to organelle acidification. Can the authors devise experiments that exclude this possibility?

2. Comparison of EXP2mRuby3 signals in Fig. 2B vs 2C shows a marked and unexpected increase in red fluorescence when C1 is removed. Comparison of panels A and B in Supplementary Fig. 1 reveals a similarly marked increase upon ML10 removal. Do the authors have an explanation for this? The Methods states “The adjustments to brightness and contrast were made only for display purposes.”, but I assume this does not account for the change in red fluorescence.

3. Along the same line, could the authors clarify how CTCF is calculated? The Methods section does not explicitly state that the total cell fluorescence is bandwidth-limited to quantify SEP fluorescence and does not include mRuby3 fluorescence. If it includes both, the data in Fig. 2D and Supp. Fig. 1C will need to be reanalyzed.

4. Lines 157-162. ML10 and C1 are not unrelated as this text implies. Both contain the diaryl-(4-piperidinyl)-pyrrole scaffold. ML10 was developed from a med-chem project based on C2, which was developed from chemistry around C1. Both compounds have very similar binding sites on PKG based on mutations in a small gatekeeper residue (PMID: 33391223). There is value in testing both compounds, but they should not be considered independent tests of the authors' model.

5. Fig. 4. Line 227 should be more cautiously worded; also at lines 441-443. Rather than being a regulated process with a specific site, it is possible that PVM rupture at one site dilutes the lytic factors, halting further digestion of the PVM at its inner face and preventing formation of a second rupture pore.

6. Supp. Fig. 3 and Movie S7. Complete loss of PMXSEP fluorescence development in imaging of cells for over 5 h upon ML10 addition contrasts with the results with C1 in Fig. 2D, where fluorescence development continued in a subset of cells blocked with C1. As stated above, PMX transcription peaks around 40-42 h (per DeRisi lab microarray data on PlasmoDB), leading me to expect fluorescence speckles when new PMX is synthesized and packaged into exonemes despite complete PKG block by ML10. To this reviewer, these data suggest that ML10 may have an off-target effect. For example, it may interfere with PMX translation and packaging into new exonemes or it might block SEP fluorescence development. Although touted to be specific, PKG mutants where the binding pocket is blocked are still killed with a growth EC50 of ~ 100 nM (see Fic. 1C of PMID: 32673324), revealing that there is some off-target toxicity associated with this compound. Can the authors perform control expts to exclude ML10 effects on PMX translation/packaging and on SEP fluorescence development?

7. The inability of bafiloymycin A1 (BaF) to increase PMXSEP fluorescence is unexpected and interpreted by the authors to implicate exoneme acidification by a mechanism other than the V type H+ ATPase pump. As this experiment was only performed in the presence of ML10, I consider this to support the above off-target effect of ML10 on SEP fluorescence development. Without excluding such an effect, the authors may want to avoid statements about mechanisms of exoneme acidification (esp. lines 389-399 but also lines 283-284).

8. Fig. 6B. These immunoblots compare ERC and PMX band intensities between PfERC-glmS/PMXSEP and PfERC-M9/PMXSEP after application of 7.5 mM GlcN for 48 h. While the M9 mutant ribozyme is a good control for glmS-mediated knockdown, it does not permit quantification of knockdown in the glmS line. Matched cultures with and without glucosamine (GlcN) treatment for the PfERC-glmS/PMXSEP are required to assess extent of ERC knockdown and to evaluate PMX processing. Such matching with the M9 mutant will also allow assessment of GlcN toxicity, which has been observed in many labs that use lower GlcN concentrations (e.g. PMID: 39829323).

Reviewer #2: My biggest concern is that the authors base all of their conclusions on live video microscopy without any biochemical data to support these claims and also that the protein selected for fusing SEP could lead to the wrong conclusions regarding the timing of events. In a recent paper by Triglia et al, (2023) it was shown that PMX has a dual localization in both exonemes and micronemes as it processes a series of proteins and proteases in both compartments. With this in mind, these results point to both exonemes and micronemes discharging their contents or have the pH changed due to an unknown mechanism or proton pump? SUB1 could have been a better target if genetic manipulation is possible and tagging does not interfere with function. Are there any other exoneme markers that could be tagged to verify the data? Regarding the timing of exocytosis, the authors state that “Another possibility is that the observed PMXSEP fluorescence at this earlier timepoint could be secretory vesicles that were not yet acidified and therefore fluorescent.” (lines 262-263). So why do the authors believe that their data point to the time of release and not lack of acidification? How can the authors exclude that these might be micronemes (yet to be acidified) where PMX resides? What is the pH of the PV? Has this ever been measured?

Additional more specific points:

1. Lines 111-112 and Figure 1E: The fact that the bright puncta are inside the PVM does not mean that this is secreted PMX in the PV. One could argue that these reside inside the merozoite. Have the authors performed any IFAs and/or expansion microscopy to show localisation of PMX at this stage? How late is that schizont and a more general comment, how do the authors distinguish between a 45h and a 47h schizont by live microscopy? Was this culture treated with C1 or ML10?

2. Lines 148-149 and Figure 2C, Movie S2: The authors state “This is followed by rapid dissolution of the fluorescence due to the rapid dilution of PMXSEP into the much larger PV compartment.” This contradicts the previous statement they made and looking at the figure and the movie, the rapid dissolution occurs after RBC rupture. In Fig.2C (panel 600 sec) the bright schizont in the middle has clearly the PVM broken. This discrepancy between the text and the Figures is even more evident in Fig.3A where intensity seems to be increased in the -30 sec panel compared to the -390 sec where the authors show PVM rupture. So PMX is clearly visible throughout the whole process of PVM rupture. Can the authors clarify their findings here? Furthermore it would have beneficial for the reader if the authors included the SEP panel in Fig.4. I appreciate the aim of this section is to study PVM rupture, however the hypothesis that it occurs at a single site using EXP2-mNeonGreen line was already been proposed and in part shown by Glushakova et al, 2019. Are there technical limitations in imaging SEP every 2 or 4 seconds? If yes, then this section should be supplementary data.

3. If the authors suggest that exoneme release occurs 3h prior to egress, that would mean that PMX and SUB1 are in the PV. Have the authors performed any Western blots with cytosolic fractions to show presence of either protease? Also this would mean that MSP1 and SERA5/6 would be processed albeit not fully if we assume that it is a slow process. A combination of IFAs and Western blots are needed to validate the above observations.

Reviewer #3: Validating the reporter system

Overall, the absence of PMX-SEP signal when PKG is inhibited and when ERC is knocked down suggests that what we see are authentic events of exocytosis of PMX-carrying exonemes. However, confirming the model of action in Fig 1A in the presented data is challenging as we are unable to visualise the vesicles fusing to the parasite's plasma membrane (as the authors themselves consider). Since this live reporter system has potential to be a great tool for studying egress, I think more controls are required for validating "tagging the pmx locus with SEP did not affect its biogenesis, maturation, function, or protein localization" and that " punctate PMX-SEP fluorescence within the PVM, indicating PMX_SEP was exposed to neutral pH and suggesting that exoneme exocytosis had occurred" (no data shown directly proves what we see are exonemes undergoing exocytosis). These could perhaps help? -

1) A PMX-GFP line that clearly shows PMX localisation in exonemes throughout irrespective of C1 treatment and whose localisation should not show the dynamics of a pH sensor.

2) IFAs to confirm exonemal and later on PV localisation of PMX-SEP. This will bridge the gap between PMX localisation data shown in previous studies (like Mukherjee et al., 2022) and this report.

Also another related suggestion - Did the authors try live imaging in the presence of E64 inhibitor that allows exonemal release but stalls the final step of RBCM rupture. This might allow us to visualise exocytosis and PVM rupture better as the merozoites will be contained within the RBC?

Presentation of live imaging data

One limitation I noted is that many of the live imaging datasets are presented as single-schizont examples, without showing the full field of view. Whilst Fig 2, supplementary videos 1 and 2 and Fig 6, supplementary videos 8 and 9 show multiple schizonts in the field, all other data are restricted to just one schizont. Since the plotted graphs presumably come from observing several schizonts within larger fields of view, it would be helpful if representative full-field videos were made available as supplementary. This would allow readers to better assess the uniformity of the reported observations. This is particularly important in relation to the discussion below on PVM rupture.

Also, minor point relating to this- for all time-lapses, would it be possible to plot normalised SEP fluorescence intensity vs time for each schizont until egress? Some videos suggest there might be some peak exocytosis activity a few secs before egress, so would be interesting to see if there is a trend there. This would also be a better representation of data shown in Fig 5B.

PVM rupture at a single site

The observation of PVM rupture at a single site in the presented videos is striking. Could the authors clarify whether this was uniformly observed across all schizonts undergoing egress within the full fields of view? Only 13 schizonts are quantified — what was the outcome for the others?

If rupture does indeed occur at a single site, why is it consistently observed at the side of the parasite? Given that parasites should be randomly oriented, one might expect some events where rupture occurs above the focal plane, following widening of that rupture, which when it reaches the focal plane of view would appear as a more uniform disappearance of the PVM. Alternatively, could the apparent single-site rupture reflect physical constraints of the setting — for instance, parasites bound to poly-lysine–coated slides or surrounded by neighbouring schizonts, which might restrict exit to one side? It is indeed puzzling that release does not appear to occur through multiple points, given that the known molecular players appear to dismantle components globally (SERA6 dismantling beta-spectrin, Thomas et al.,2018).

It may be helpful if the authors could clarify their use of terms ‘rupture’ and ‘poration’, as they appear to be used somewhat interchangeably. These could represent distinct steps. Prior work has suggested that PVM rounding may be associated with or a consequence of poration that precedes PVM rupture (Tan & Blackman, 2021). It is not entirely clear whether the observed break in EXP2-mRuby signal reflects a single opening (pore) or the focal plane view of a broader rupture event.

**Part III – Minor Issues: Editorial and Data Presentation Modifications**

Reviewer #1: 1. Abstract (line 24) and Introduction (lines 82-83) are vague about the second “fluorescent marker on the PVM”. This reviewer was left confused and wondering. I suggest describing that this as c-terminal tagging of EXP2 with mRuby or a GFP derivative so that readers can better anticipate how PVM rupture was tracked.

2. “However, the spatiotemporal regulation of exocytosis, PVM breakdown, and merozoite egress remains unknown.” This statement is a bit unfair as there is much known about exoneme discharge, timing of SUB1-mediated proteolysis of targets, subsequent PVM breakdown and egress, all occurring within ~ 8 min. Even if the authors perform control experiments and properly establish that occasional discharge of a few exonemes begins as early as 3 h before merozoite egress, their data suggests (and I assume they would agree) that the majority of exonemes are discharged shortly before the 8 min cascade of events that culminate in merozoite egress. Isn’t it better to assume that rare exoneme discharge events occurring up to 3 h before egress are spurious cellular events and that quantitative discharge is needed to achieve the required threshold processing of SUB1 targets.

3. Lines 181-82. Ref. 12 is a review and might not be the most appropriate for statements about the spatio-temporal relationship between exoneme exocytosis and merozoite egress. PMID: 28683142 uses transfectant clones with conditional KO of SERA genes, Western blotting for activation of SUB1 proteolytic activity against its targets, kinetics of egress in living parasites after washout of C1 and C2 to implicate rapid egress (< 8 min) upon washout of PKG inhibitors. Thus, it seems a bit unfair to say that the kinetics of exoneme exocytosis and subsequent egress have only been studied with fixed parasites.

4. Lines 150-151: “These data suggest that exoneme exocytosis occurs all over the schizont and may not be limited to a single exit site or specific cellular location within the merozoite.” As there are multiple merozoites within a schizont and there is significant Brownian/active movement of merozoites in live recordings, this statement should be removed or significantly toned down.

5. Please revise Lines 392-393. Ref. 31 applied BaF for only 2 h and did not examine egress vs. invasion specifically as they examined only the result with Giemsa staining after 6 h. Also, please correct this citation from BioRxiv to JBC as this paper has been published for some time - PMID: 39084459.

Reviewer #2: 1. Line EXP2mRuby3/PMXSEP is clonal and seems to grow normally but a growth curve should be shown (supplementary data). Also was ATc added after transfection for a brief time or was it present throughout culturing? The authors briefly mention presence of ATc but this should be explained in more detail as one would assume that presence or absence of ATc even briefly might affect PMX levels?

2. Genotyping PCRs showing absence of the endogenous locus in line EXP2mRuby3/PMXSEP should be provided as supplementary material.

3. Line 304: “To test this hypothesis, we first generated PfERC conditional mutants using the glucosamine (GlcN) responsive ribozyme system (Fig. 6A)”. The authors have already generated that line for their previous publication focusing on PfERC. Is this a new one? Is the genetic background different? Please clarify.

4. Line 322-323: “The addition of GlcN to PfERC-glmS/PMXSEP and PfERC-M9/PMXSEP parasites also resulted in loss of PMX processing (Fig. 6B).” One would expect that addition of GlcN to PfERC-M9/PMXSEP would have no effect on PMX processing as PfERC levels remain unaffected. Can the authors clarify this?

5. Figure 7, lines 460-471: Although I agree with the concept of two independent signals for egress, irrelevant of the timing, there are data that suggest that exoneme exocytosis might not be Calcium dependent (Balestra et al, 2021) and Calcium flux is correlated only with microneme release. One would think that if there are 2 calcium fluxes even from distinct stores they would have probably been observed in studies with Fluo4 or Calcium reporters that have been used in Plasmodium, unless they occur at the same time. Can the authors elaborate on their rational for this model?

Reviewer #3: Fig 2 A - please mention total number of schizonts from which percentages were calculated

Fig 5 should also have the C1 panel for comparison.

Fig 6C and movies 8 and 9. - Has the fluorescence channel been merged with DIC? could the background speckles be reduced for better clarity?

"The data shows" and "These data show" are a bit overused throughout the manuscript. Perhaps omit saying this.

A lot of statements are repeated in the Introduction, Results and Discussion. Perhaps reducing this would make the text more crisp and clear.

Line 70-79 seems out of place here. Could be omitted in the Intro and appear only when relevant in the Results when talking about ERC.

Typos - "Intergrants" in Fig 1B

PLOS authors have the option to publish the peer review history of their article (what does this mean?). If published, this will include your full peer review and any attached files.

Reviewer #1: No

Reviewer #2: No

Reviewer #3: **Yes:**Abhinay Ramaprasad

**Figure resubmission:**
---

## [Decision Letter · Decision Letter 1]

1 Feb 2026

PPATHOGENS-D-25-02207R1

Spatiotemporal dynamics of signal dependent exocytosis and parasitophorous vacuolar membrane rupture during Plasmodium falciparum egress.

PLOS Pathogens

Dear Dr. Muralidharan,

Thank you for submitting your manuscript to PLOS Pathogens. After careful consideration, we feel that it has merit but does not fully meet PLOS Pathogens's publication criteria as it currently stands. Therefore, we invite you to submit a revised version of the manuscript that addresses the points raised during the review process.

We look forward to receiving your revised manuscript.

Kind regards,

Mathieu Brochet

Academic Editor

PLOS Pathogens

Dominique Soldati-Favre

Section Editor

PLOS Pathogens

Sumita Bhaduri-McIntosh

Editor-in-Chief

PLOS Pathogens

orcid.org/0000-0003-2946-9497

Michael Malim

Editor-in-Chief

PLOS Pathogens

orcid.org/0000-0002-7699-2064

**Additional Editor Comments:**

Your manuscript has been re-evaluated by the same three reviewers, who acknowledge that it has substantially improved. Nevertheless, reviewers 1 and 2 remain unconvinced that the PMX-SEP fluorescent signal observed upon packaging into exonemes cannot originate from the cytosol, nor that secretion is definitively demonstrated. They therefore request additional experimental evidence to directly address these points. While such experiments may present technical challenges, they would be essential to rule out alternative interpretations of the data and to robustly support the proposed model. Accordingly, we recommend strengthening the manuscript by providing additional experimental validation of the model presented in Figure 1A.

**Journal Requirements:**

1) Please provide an Author Summary. This should appear in your manuscript between the Abstract (if applicable) and the Introduction, and should be 150-200 words long. The aim should be to make your findings accessible to a wide audience that includes both scientists and non-scientists. Sample summaries can be found on our website under Submission Guidelines:

https://journals.plos.org/plospathogens/s/submission-guidelines#loc-parts-of-a-submission

- ® on page: 22.

3) We notice that your supplementary Tables are included in the manuscript file. Please remove them and upload them with the file type 'Supporting Information'. Please ensure that each Supporting Information file has a legend listed in the manuscript after the references list.

Potential Copyright Issues:

i) Figures 1A, 6A, and 7. Please confirm whether you drew the images / clip-art within the figure panels by hand. If you did not draw the images, please provide (a) a link to the source of the images or icons and their license / terms of use; or (b) written permission from the copyright holder to publish the images or icons under our CC BY 4.0 license. Alternatively, you may replace the images with open source alternatives. See these open source resources you may use to replace images / clip-art:

**Reviewers' Comments:**

Reviewer's Responses to Questions

**Part I - Summary**

Reviewer #1: The authors have done an excellent job addressing all raised concerns. The addition of NH4Cl treatment to identify acidified exonemes prior to discharge is an elegant control. The conclusions have been toned down appropriately. The manuscript will add significantly to our understanding of merozoite maturation, PVM rupture, and egress, events that are important targets for therapy development.

Reviewer #2: Dedkhad et al, provided an updated version of the manuscript which includes some additional data regarding the timing of exoneme exocytosis. However, the authors did not address many of my concerns experimentally and although the technical challenges are substantial the data presented only partially support the claims made. With the use of inhibitors (which probably do not reflect what happens in real time) this is a process that can be completed in 15 minutes, but the authors suggest that the parasite under normal conditions does it in 3h. Although I find the model very exciting clearly pointing to diverse signalling events to regulate every process, I respectfully disagree with the authors that detailed characterisation on what the fluorescent signals are should be the focus of future studies.

Reviewer #3: I thank the authors for considering the issues raised and addressing them with more data and clarifications.

**Part II – Major Issues: Key Experiments Required for Acceptance**

Reviewer #1: (No Response)

Reviewer #2: My main concern still remains that the authors cannot provide data to distinguish if the fluorescent puncta is PMX secreted in the PV or PMX in non-acidified exonemes/micronemes. Can the authors exclude the possibility that this acidic organelles via a PKG regulated mechanism change their pH? The ammonium chloride experiments show that these compartments become fluorescent (i.e with neutral pH). Does this suggest exoneme secretion? If not, the parasite could trigger de-acidification in a PKG dependent manner to stop PMX activity as prolonged protease activity could lead to non-specific cleavage.

Regarding the localisation of PMX, Mukherjee at al (Nat. Commun. 2022) analysed in detail via EM colocalisation of PMX and SUB1 with AMA1 and rhopty markers, however they only speculated about a "precursor compartment" as no data were provided for that. Distribution of these proteins and potential overlap in these organelles is still under investigation.

The authors state that they do not have any PMX or SUB1, nor MSP1 or SERA5 antibodies, which I agree makes it very difficult to study some of the events in detail, although they have used these antibodies in the past. However, they do have a PMX-V5 tagged line where they could have looked for PV localised PMX in a time course experiment either by WB or IFA.

Reviewer #3: I appreciate the limitations posed by non-availability of valid exonemal markers to validate their system and to confirm correct subcellular localisation of PMX-SEP. However, I still think the system would greatly benefit from direct validation of the model in Figure 1A, to confirm that the fluorescence we see in the videos are triggered by a change in pH specifically upon exoneme exocytosis. While the authors say the PMX-SEP is non-fluorescent until exocytosis, and that a punctate pattern is observed within the enclosed PVM, none of the videos rule out cytosolic origin of these signals when PMX is packaged into exonemes nor do they effectively show localisation to the PPM. So, either a non-pH sensing PMX-tag control (in which we would see a contrasting, more stable signal throughout compared to PMX-SEP, and possibly reacts differently to C1 treatment and ERC knockdown) or any other form of similar control experiment would help prove these are authentic exocytosis events, clarify certain questions about the timing of exocytosis and would further bolster the paper in my opinion. On the other hand, since treatments with C1, ammonium chloride and ERC knockdown do have an effect on PMX-SEP that supports their model, it is reasonable to say that the evidence supports the conclusions as of now.

To clarify my point about E64 - in videos such as Movie 3, we cease to see PMX-SEP and EXP2-mRuby signals upon egress because the cell has burst and merozoites dispersed. E64 might allow us more time to observe these signals after exoneme release has been triggered. Do merozoites continue exoneme release after practically been "released" from the PV? However, I agree with the authors that it might not add significant more detail.

The authors have satisfactorily resolved other issues by adding more videos and schizont numbers to support their conclusions.

**Part III – Minor Issues: Editorial and Data Presentation Modifications**

Reviewer #1: (No Response)

Reviewer #2: 1. Regarding the timing of exocytosis, the authors state that “Another possibility is that the

observed PMXSEP fluorescence at this earlier timepoint could be secretory vesicles that were

not yet acidified and therefore fluorescent.” (lines 262-263).

Author response: Newly included data using ammonium chloride is inconsistent with this

hypothesis. Therefore, we have removed this line from the manuscript.

This has not been removed from the revised manuscript (lines 259-260)

2. Figure legend for 2E and 2F is missing

3. For the ammonium chloride experiments, the authors reference Lopes da Silva M et al. in their rebuttal letter. In that study Ammonium chloride was used for disruption of host liver cell acidification, which is not mentioned. Furthermore I could not find that reference in the revised manuscript, yet the authors reference Shadija N et al, and I could not find in which experiment they used Ammonium chloride.

4.Line 296 "Calcium signals are known to be essential for exoneme exocytosis". That is not the case. "They might be important" is more accurate.

5. Since the proteases essential for egress are activated by low pH[15,35–38] (lines 386-387). SUB1 is active at neutral pH

Reviewer #3: (No Response)

PLOS authors have the option to publish the peer review history of their article (what does this mean?). If published, this will include your full peer review and any attached files.

Reviewer #1: No

Reviewer #2: No

Reviewer #3: **Yes:**Abhinay Ramaprasad

**Figure resubmission:**
---

## [Editor Report · Decision Letter 2]

29 Apr 2026

Dear Dr. Muralidharan,

We are pleased to inform you that your manuscript 'Spatiotemporal dynamics of signal dependent exocytosis and parasitophorous vacuolar membrane rupture during Plasmodium falciparum egress.' has been provisionally accepted for publication in PLOS Pathogens.

Best regards,

Mathieu Brochet

Academic Editor

PLOS Pathogens

Dominique Soldati-Favre

Section Editor

PLOS Pathogens

Sumita Bhaduri-McIntosh

Editor-in-Chief

PLOS Pathogens

orcid.org/0000-0003-2946-9497

Michael Malim

Editor-in-Chief

PLOS Pathogens

orcid.org/0000-0002-7699-2064
---

## [Editor Report · Acceptance letter]

Dear Dr. Muralidharan,

We are delighted to inform you that your manuscript, "Spatiotemporal dynamics of signal dependent exocytosis and parasitophorous vacuolar membrane rupture during Plasmodium falciparum egress.," has been formally accepted for publication in PLOS Pathogens.

Best regards,

Sumita Bhaduri-McIntosh

Editor-in-Chief

PLOS Pathogens

orcid.org/0000-0003-2946-9497

Michael Malim

Editor-in-Chief

PLOS Pathogens

orcid.org/0000-0002-7699-2064